# Modelled fracture and calving on the Totten Ice Shelf

Sue Cook[1], Jan Åström[2], Thomas Zwinger[2], Benjamin Keith Galton-Fenzi[3,1], Jamin Stevens Greenbaum[4], Richard Coleman[5,1]

[1] Antarctic Climate & Ecosystems Cooperative Research Centre, University of Tasmania, Hobart, Tasmania 7001, Australia.

[2] CSC - IT Center for Science, FI-02101 Espoo, Finland.

[3] Australian Antarctic Division, Kingston, Tasmania 7050, Australia.

[4] Institute for Geophysics, University of Texas at Austin, Austin, TX 78758, USA.

[5] Institute for Marine and Antarctic Studies, University of Tasmania, Hobart, Tasmania 7001, Australia.

*Correspondence to*: Sue Cook (sue.cook@utas.edu.au)

**Abstract.** The Totten Ice Shelf (IS) has a large drainage basin, much of which is grounded below sea level, leaving the glacier vulnerable to retreat through the Marine Ice Shelf Instability mechanism. The ice shelf has also been shown to be sensitive to changes in calving rate, as a very small retreat of the calving front from its current position is predicted to cause a change in flow at the grounding line. Therefore understanding the processes behind calving on the Totten IS is key to predicting its future sea level rise contribution. Here we use the Helsinki Discrete Element Model (HiDEM) to show that not all of the fractures visible at the front of the Totten IS are produced locally, but that the across-flow basal crevasses, which are part of the distinctive crosscutting fracture pattern, are advected into the calving front area from upstream. A separate simulation of the grounding line shows that re-grounding points may be key areas of basal crevasse production, and can produce basal crevasses in both an along and across flow orientation. The along-flow basal crevasses at the grounding line may be a possible precursor to basal channels, while we suggest the across-flow grounding line fractures are the source of the across-flow features observed at the calving front. We use two additional models to simulate the evolution of basal fractures as they advect downstream, demonstrating that both strain and ocean melt have the potential to deform narrow fractures into the broad basal features observed near the calving front. The wide range of factors which influence fracture patterns and calving on this glacier will be a challenge for predicting its future mass loss.

# 1 Introduction

The calving of icebergs is responsible for around half of the mass lost from the Antarctic ice sheet (Depoorter et al., 2013; Liu et al., 2015; Rignot et al., 2013). Despite this, the calving process is not currently well represented in the large scale ice sheet models, with different calving approaches producing widely varying predictions of Antarctica's future sea level rise contribution (e.g. Deconto and Pollard, 2016; Levermann et al., 2014). Calving is ultimately controlled by how fractures develop on a glacier, and the key processes involved can range from lateral rift propagation in the case of large icebergs, or the vertical penetration of a dense crevasse field in ice shelf disintegration (Benn et al., 2007). Fractures involved in calving may form in situ, or be advected into the calving front region from upstream. Basal crevasses have been observed near the grounding line of ice shelves (e.g. Bindschadler et al., 2011b; Jacobel et al., 2014) and may evolve in shape and size as they advect downstream (Jordan et al., 2014). To fully understand the calving behaviour of an ice shelf, we must therefore understand how fractures form and propagate on it. In this paper we apply fracture modelling to the Totten Ice Shelf (IS) to investigate these processes.

The Totten IS is fed by a large drainage basin, containing at least 3.5 m global sea level equivalent (Greenbaum et al., 2015). This region of the Antarctic Ice Sheet is grounded largely below sea level (Greenbaum et al., 2015), and has shown signs of significant grounding line fluctuation in the past (Aitken et al., 2016), leading to speculation that it may be vulnerable to Marine Ice Sheet Instability (Greenbaum et al., 2015; Pollard et al., 2015; Weertman, 1974). There have been some indications that the Totten IS may already have been thinning (Khazendar et al., 2013), although it is unclear if this represents a long term trend (Paolo et al., 2015). Crucially, the Totten IS has been identified as having one of the lowest proportions of "passive" ice in Antarctica (Fürst et al., 2016), i.e. the proportion of the ice shelf which can be lost without a resulting dynamic change in the grounded ice upstream. This means that any change in calving rate on the Totten IS could quickly cause an increased flux of ice across its grounding line and hence an increased contribution to global sea level rise.

The Totten IS has a dense pattern of both longitudinal and transversal fractures near the calving front, typical of fast flowing ice shelves in Antarctica, resulting in the frequent release of small tabular icebergs (Wesche et al., 2013). Little is currently known about what fractures exist upstream of the calving front, as these are highly challenging to observe from satellite imagery. High rates of snowfall on the Totten IS act to quickly fill any surface crevasses, while the thickness of the ice shelf

means that basal crevasses would need to be extremely large to have an observable surface expression. However, the water in the ocean cavity underneath the Totten IS is known to be warm enough to drive melt of the ice (Gwyther et al., 2014; Rintoul et al., 2016; Roberts et al., 2018). Therefore, any basal crevasses exposed to seawater are likely to be affected by melting.

The first stage to predicting future calving rates from the Totten IS is to be able to reproduce current fracture patterns and calving rates. One tool which we can use to improve our understanding of the processes behind fracture and calving on ice shelves is a discrete element model (DEM). A DEM discretises the ice shelf geometry into a lattice of bonded particles. By allowing the bonds between particles to break, the model is able to explicitly develop fractures, rather than representing fracturing by a homogenized variable as in continuum approaches. Discrete element models have been used successfully to simulate calving behaviour at marine terminating glaciers (Bassis and Jacobs, 2013; Vallot et al., 2017), to examine causes behind changes in calving behaviour (Åström et al., 2014) and to investigate how fracture propagation is affected by other elements of the cryospheric system (Benn et al., 2017; Vallot et al., 2017).

In this paper we present the first application of a discrete element model to an Antarctic ice shelf, and use it to examine fracture and calving on the Totten IS. The DEM can only be used to simulate small regions of an ice shelf due to its high computational cost. We choose to apply the model to two sections of the ice shelf most likely to experience fracture development: the calving front, and the grounding line. The DEM results at the calving front demonstrate that the observed transverse fractures cannot be reproduced locally, and therefore must advect into the region from upstream. The grounding line model demonstrates that across-flow fractures can be produced at pinning points near the grounding line. We then use two additional models to investigate how fractures produced at the grounding line might be modified as they advect downstream: a continuum ice-flow model to simulate the effects of internal strain, and the Kardar-Parisi-Zhang (KPZ) equation to simulate fracture development under a constant marine melting. These results are used to demonstrate that the observed fracture pattern at the calving front can plausibly be explained by a combination of both locally produced fractures and basal features produced initially at the grounding line. In combination, the three models produce a description of the key areas of fracture production on the Totten IS.

**2 Data and Methods**

## 2.1 Glaciological Data

The geometry of the discrete element model is taken from aerogeophysical data collected by the International Collaboration for Exploration of the Cryosphere through Aerogeophysical Profiling (ICECAP) project, which has flown a dense network of flights over the Totten IS. The surface elevation (Blankenship et al., 2015) and ice thickness (Blankenship et al., 2012) data are interpolated using TELVIS (Thickness Estimation by a Lagrangian Validated Interpolation Scheme) (Roberts et al., 2011), to produce a geometry which is glaciologically self-consistent and has previously been used to infer basal melt rates on the glacier (Roberts et al., 2018). The bedrock elevation, which is used to distinguish zones of grounded/floating ice, is taken from gravity inversions (Greenbaum et al., 2015). Elevation and thickness data from an individual flight path from the ICECAP campaign are also used to assess the output of the KPZ equation, and to extract more information on the types of basal fractures observed on the ice shelf.

The performance of the discrete element model is evaluated by comparing the size and frequency of simulated icebergs with those observed in satellite images. The observed iceberg size distribution is calculated from three separate Landsat 7 images from 2009, 2010 and 2011, with each iceberg within 20 km of the calving front mapped and its area measured. The distance is chosen to ensure that only icebergs originating from the Totten IS are included. The three images produce a sample of 370 observed icebergs, and cover periods of both low and high iceberg production rates to produce a representative sample (see Supplementary Material).

## 2.2 Discrete Element Model

We model fracture formation on the Totten IS using the Helsinki Discrete Element Model (HiDEM) which is described in detail in Åström et al. (2013) and Riikilä et al. (2014). The model geometry is constructed with interconnected inelastic blocks, each representing a discrete volume of ice. The blocks are connected to their neighbours by breakable elastic beams whose energy dissipation depends on the deformation rate. As the ice deforms under its own weight, stresses on the beams increase; if stress reaches a failure threshold, the beam breaks creating a fracture in the modelled ice shelf. We have chosen a critical stress of approximately 1 MPa, governed by previous applications of the model to glacier ice (Åström et al., 2013, 2014). The range of possible yield stresses that produce plausible results is not large, and previous results indicate that changing yield stress has the effect of changing the number of fractures but not their underlying orientation. Therefore the

assessment of our results is restricted to orientation of fractures rather than their density.

The movement of each block of ice is calculated using a discrete version of Newton's equation of motion and is iterated with a timestep of 0.0001 s length. For computational reasons, we use a densely packed, face-centred cubic (fcc) lattice of spherical blocks. The shape of the lattice introduces a weak directional bias in the elastic and fracture properties of the ice.

The symmetry of the underlying fcc-lattice is however easily broken by the propagating cracks, as evidenced by the results. At the beginning of a fracture simulation, the ice is assumed to contain a low density of randomly scattered small pre-existing cracks. For these experiments, approximately 1% of the bonds are broken at the beginning of the simulation.

The effect of the surrounding ocean is simulated by applying a buoyancy force ($F_b$) to each block of ice forming the model domain:

$$F_b = (\rho_w - \rho_i)gV_p ,\qquad(1)$$

where $\rho_w$ and $\rho_i$ are the densities of ocean water and ice respectively, $g$ is the gravitational constant and $V_p$ is the volume of a block. Since the model domain cannot cover the whole of the ice shelf due to computational cost, some of the boundaries of the domain are necessarily not simply floating passively, but are being pushed by the up-stream pressure from the land-based flowing part of the glacier. This pressure was modelled by applying isostatic pressure, $P_{sc}$, at the lateral boundaries of

the simulated area where required.

$$P_{sc} = \begin{cases} \rho_i\, g\, z & z > 0 \\ (\rho_i - \rho_w)\, g\, z & z < 0 \end{cases},\qquad(2)$$

where z is the elevation above sea level. This is a simplification of the driving stress, as it neglects longitudinal stresses which can be significant around the grounding line (Schoof, 2007). Where the model domain is grounded a basal shear stress $F_f$ is applied:

$$F_f = -c\frac{dx}{dt},\qquad(3)$$

where $\frac{dx}{dt}$ is the sliding velocity and c is a drag coefficient set at $10^7$ kg m$^{-2}$ s$^{-1}$.

The model is applied to two regions of the ice shelf, one at the calving front and one around the southern tip of the grounding

line. Each model setup uses the observed geometry described above. Our first model domain is 55 x 40 km covering roughly 2000 km² of the Totten IS calving front (Fig. 1, 2). The area of the ice shelf covered in the calving front domain is almost entirely floating. A particle size of 50 m was chosen, creating a model domain of approximately 5 million individual particles and a minimum stack of 5 particles vertically. Our second model domain covers a 45 x 45 km section of ice at the

southern-most point of the Totten IS (Fig. 1, 3). For this model a 150 m particle size was used, creating a model set up with 2.5 million particles and a stack of between 7 and 19 particles vertically. Lastly, the model is run again using the calving front domain, but this time a series of basal crevasses are artificially introduced into the geometry mimicking the observed basal profile with straight, across-flow basal features with a height $h$ given by a trigonometric function:

$$h = 250 \sin(0.005x)^8 \cos(0.015x)^2 \qquad . \tag{4}$$

This produces regular basal features with spacing of around 600 m, height around 100 m and width of around 300 m, designed to roughly mimic the presence of basal crevasses. Since the model domain was chosen to align with the dominant flow direction to improve computational efficiency, the introduced features also align with the model grid. This model run is used to simulate the effect of fractures which were advected into the domain rather than produced locally.

## 2.3 Kardar-Parisi-Zhang (KPZ) Equation

To investigate how basal crevasses might evolve over time, we use two models to simulate the effects of basal melt and ice deformation separately. The first model simulates widening of basal cracks by ocean-driven melt in a simplistic fashion, by describing the melting ice base as a propagating front. If melting is assumed spatially constant, the ice base will melt, as a first approximation, along the upward normal of its local surface. A generic model for such a front is the stochastic non-linear differential equation called the KPZ-equation (Kardar et al., 1986). The KPZ equation has been extensively used to describe,

for example, slow combustion fronts or the growth of bacterial colonies (e.g. Bonachela et al., 2011; Lam et al., 2017). By assuming a constant ice velocity and using a Lagrangian coordinate system we can neglect the movement of the physical interface between the ice shelf and ocean caused by velocity of the ice. This allows us to neglect advection terms and to treat the interface as stationary. The 1-dimensional KPZ equation then reads:

$$\frac{\partial h(x',t)}{\partial t} = \nu \nabla^2 h(x',t) + \frac{\lambda}{2} [\nabla h(x',t)]^2 + F + \eta(x',t) \qquad , \tag{5}$$

where the vertical front location $h$ varies with Lagrangian coordinate $x'$ and time $t$, assuming a constant along flow velocity of 1 km a$^{-1}$. This is a simplification, as the ice shelf does experience along flow acceleration, but the method provides a first order estimate of the effects of melting excluding strain thinning. The first term on the r.h.s. can be thought of as a 'bending stiffness' that keeps the front from fluctuating too much, the second term drives the front perpendicular to its normal, $F$ is the

driving force (melting) and $\eta$ is a stochastic term which can be used to introduce noise, but in this case we use it to represent a series of initial basal crevasses to which the propagation of the front is applied.

To investigate the role of ocean melting we use the KPZ equation applied to a set of line cracks in an idealised and homogeneous 1 km thick floating ice shelf (Fig. 4). The vertical red line segments at the bottom of the figure represent initial line cracks, represented by $\eta$ in Equation 4. The melting front is then propagated forward in time with $\nu = 1$ a$^{-1}$, $\lambda = 1$ m a$^{-1}$

and $\eta(t>0) = 0$. To represent the longer exposure to melt for downstream ice we linearly increase the driving force $F$ along the flowline: $F = 1x10^{-3}x'$ m a$^{-1}$. The initial crack configuration is varied until a solution is found that approximates the measured base along the flight path. The surface elevation is thereafter calculated by assuming neutral buoyancy locally.

The KPZ approach used here makes a number of significant simplifications in modelling basal melt. The approach neglects spatial variations in melt rate across the ice shelf, which can be significant (e. g. Gwyther et al., 2014; Roberts et al., 2018).

More importantly, the constant melt rate used neglects variations in melt rate which have been predicted to occur within basal features, where melt rates at the peak of a basal crevasse are typically found to be lower than on the walls (Jordan et al., 2014; Khazendar and Jenkins, 2003; Millgate et al., 2013). This internal difference in melt rates would change the shape of the basal crevasses as they evolve.

**2.4 Elmer/Ice**

The effects of internal ice deformation will also change the shape of a basal fracture as it advects along with the ice shelf. To simulate how this may work we use the finite element model Elmer/Ice (Gagliardini et al., 2013) to simulate the changing shape of an idealised basal crevasse in two dimensions (2D). The model uses an idealised domain with an initial narrow basal crevasse in a linear slab of ice, similar to the geometry used for the KPZ equation but this time with a single crack.

The shelf is represented by a slab of ice initially 680 m thick and 4 km long, floating with neutral buoyancy, with an imposed

basal crevasse 200 m wide at the base which penetrates 300 m vertically into the shelf (Fig. 5). Longitudinal stretching is simulated by fixing the left hand boundary of the domain, while moving the right hand side at 500 m a$^{-1}$. This is roughly equivalent to the maximum rate of stretching experienced on the Totten IS at the grounding line. The domain is iterated forward in time with timesteps of 12 hr, with the 2D velocity field solved at each time step using the Stokes equation

together with Glen's flow law with an exponent of n=3 (see Gagliardini et al. (2013) for more detail). We choose a constant temperature within the ice of -9°C and an enhancement factor of one. The geometry of the domain is evolved with a kinematic condition for the free surface at the top and bottom boundaries, and the model is automatically adjusted to remain in hydrostatic equilibrium by maintaining the lower boundary at seawater pressure.

**3 Results**

The HiDEM model was run on two domains, one at the calving front and one near the grounding line. Our first model setup covers the Totten IS calving front, with almost all of the domain consisting of floating ice (Fig. 1, 2). The modelled fracture pattern shows tensile cracks, with the majority running roughly perpendicular to the calving front or parallel to ice flow and cutting through the full ice thickness (Fig. 2b). These cracks appear where the calving front begins to spread laterally, as it is pushed towards the open sea by the upstream pressure. Other modelled fractures at the eastern edge of the domain are caused

by shear around a pinning point and form at an angle to the flow direction. Satellite observations of fracture patterns are hampered by thick snow cover on the Totten IS, however near the calving front similar cracks perpendicular to the calving front are clearly visible in satellite imagery (Fig. 2a). The satellite imagery also shows a number of across-flow features, which can be observed tens of kilometres upstream of the calving front (Fig. 2a). These features are similar in form to those observed on Larsen C, which have been shown to be the surface expression of large basal crevasses (Luckman et al., 2012). Since these

across-flow basal features are not reproduced by the HiDEM model we infer that they are advected into the calving front region from further upstream.

Our second HiDEM domain covers the southern-most point of the Totten IS (Fig. 1, 3). The observed geometry in this region shows a grounding zone rather than a distinct grounding line, where the ice re-grounds on many small islands (Fig. 3). In this region, the HiDEM simulation produces a number of fractures at the base of the ice shelf, which cluster around the re-

grounding islands (Fig. 3). The basal crevasses have a tendency to form either in the along-flow direction as shear cracks or in the across-flow direction as tensile cracks. The crevasses typically reach 200 m above the base of the ice.

The fractures produced by the HiDEM model at the grounding line are narrow (less than 100 m wide at the base). Examining geometry data from the ICECAP survey, the across-flow basal crevasses causing a visible surface expression are much larger with a width of 1 – 3 km and height of 150 – 300 m (Fig. 4). To investigate whether the narrow basal crevasses formed at the grounding line might evolve into these broad features as they move downstream, we use two models to simulate the effects of basal melting and internal strain separately.

The first experiment uses the KPZ equation to apply constant basal melting to a set of narrow basal fractures, selected with positions and heights specifically designed to match the observed geometry along a single ICECAP flightline (Fig. 4). The model was run forward over a period of 100 years, which is approximately equal to the transit time from grounding line to calving front. In this way it is possible to reproduce both the surface and basal features of the Totten IS on an along-flow flight path (Fig. 4). This demonstrates that basal melting is a plausible mechanism for turning narrow basal crevasses at the grounding line into the broader basal features observed near the calving front.

We separately investigate how internal ice deformation may affect the shape of basal crevasses using an idealised simulation of a single basal crevasse in Elmer/Ice. As the model runs forward over time, longitudinal strain causes the initial crack to widen from 200 to 350 m after 52 weeks (Fig. 5). At the same time the height of the crevasse decreases from 300 m to 180 m. This simple pair of model experiments demonstrate that narrow fractures produced at the grounding line are a possible source for the wider basal features observed near the calving front. Both basal melting and internal strain can act to widen an initially narrow basal crevasse, although only the basal melt experiment is able to maintain a large vertical extent.

Lastly, to investigate the importance of these advected, across-flow features in the calving front region we perform an additional HiDEM simulation of the calving front, with across-flow basal crevasses artificially introduced into the geometry. In this model realization, the Totten IS begins to form visible full-thickness fractures in both the along- and across-flow directions (Fig. 6).

An additional method for examining the model output is to compare the fragment size distribution (FSD) of the icebergs produced. The form of the FSD is controlled by the type of fracture mechanism occurring in the model. An FSD with the form "area$^{-3/2}$" indicates a dominant mechanism of brittle fragmentation, while distributions with a higher slope indicate that grinding and crushing of fragments is occurring (Åström, 2006). A slope of -3/2 has previously been observed for Antarctic icebergs (Tournadre et al., 2016), and is also observed for icebergs produced by Totten IS, although smaller icebergs are under-represented due to systematic difficulties in optically resolving small icebergs and distinguishing smaller icebergs from sea ice (see Supplementary Material). Both HiDEM simulations (with and without introduced basal crevasses) reproduce the observed distribution of "area$^{-3/2}$" at large iceberg sizes, which indicates that the models accurately reproduce the brittle fragmentation process driving large iceberg production. At smaller iceberg sizes, the simulations show a higher slope, indicating that grinding of smaller fragments occurs in the simulations (Fig.7).

## 4 Discussion

The initial HiDEM simulation of fracture development at the calving front of the Totten IS replicates the observed pattern of full thickness fractures produced roughly perpendicular to the calving front, but cannot reproduce the observed basal crevasses running across-flow behind the calving front. These features appear to advect into the calving front area from further upstream, a conclusion which is supported by their wide and rounded shape (Fig. 4). Recently produced fractures would be more likely to have a narrower and sharper shape, so the broad and round shapes observed indicate that other processes such as strain deformation or melting have modified the crevasse's shape over time. We cannot completely rule out the possibility of across-flow fractures being produced in the calving front area, since our model uses a simplified boundary condition where the driving stress is approximated by the isostatic pressure. This neglects longitudinal stresses, which can be significant around grounding lines and transition zones (Schoof, 2007). Were these stresses to be included in the model boundary conditions, it is possible that the fracture pattern would be altered.

We hypothesise that the across-flow basal features originate at the grounding line, as basal crevassing at the grounding line has previously been observed on other ice shelves (e.g. Jacobel et al., 2014). A second HiDEM experiment around the southern-most grounding zone shows that high levels of basal crevassing occur particularly near to pinning points. Here the upward gradient in the bedrock causes compression of the ice, and conversely extension on the top and lee-side of the island. It is this

process that causes the basal fractures. If these fractures were caused by a sharp gradient in basal shear stress as the ice transitions from grounded to floating, then fractures would be expected to occur at the first point of ungrounding as well as at pinning points. Instead, we see fractures occurring primarily not at the initial point of floatation, but at re-grounding islands. If the drag coefficient were increased then basal crevasses might begin to appear at the initial grounding line also, and basal

crevasse production may thus depend on the subglacial environment at the grounding line.

The model predicts the formation of both across- and along-flow basal crevasses in the grounding zone, with the along-flow crevasses caused by shear between the freely floating ice and that grounded on the pinning points. The across-flow features are hypothesised to be the source of across-flow surface features observed downstream on the Totten IS. It is possible that along-flow basal crevasses could be a nucleus for the formation of basal channels, which subsequently grow by enhanced

melting (Drews, 2015). This is a potential mechanism for the development of grounding-line sourced channels, the cause of which can be unclear in the absence of obvious subglacial hydrology (Alley et al., 2016).

As the across-flow basal crevasses advect downstream, ocean-driven melt and lateral spreading of the ice are likely to widen them, until they are large enough near the calving front to produce a visible surface expression. Basal melt rates are highest at the southern end of the Totten Ice Shelf (Gwyther et al., 2014) where the thicker ice shelf means that the pressure dependent

melting point is reduced at the base. Nearer the calving front, accelerating velocities and lateral spreading of the ice means internal deformation is more likely to be a dominant factor in basal crevasse evolution (e.g. Li et al., 2016).

We have modelled the two effects separately, using the KPZ to simulate a spatially constant basal melt rate and a full-Stokes ice flow model (Elmer/Ice) to examine the effect of longitudinal stretching on basal crevasse shape. Using the KPZ equation to model the development of fractures under ocean melting, we demonstrate that broad across-flow basal features observed in

aerogeophysical data can be reproduced by applying a spatially constant melt rate. Likewise, our Elmer/Ice simulation was able to produce a broadening of an initial narrow basal crevasse through longitudinal stretching. Both models make significant simplifications by assuming no spatial or temporal variation in conditions, but demonstrate that narrow features at the grounding line are a plausible source for the broader features observed near the calving front of the Totten Glacier, as has previously been suggested by Bassis and Ma (2015). It should also be stated that basal crevasses have also been observed to

form downstream of the grounding line (Luckman et al., 2012) and we cannot rule out the possibility that further basal fractures

form between the grounding line and the calving front. The widening of basal crevasses by longitudinal stretching also depends on the strain rate, and for ice shelves with a lower strain rate basal crevasses could in fact close as they advect downstream.

The simulation of longitudinal stretching was able to widen the initial crevasse, but the height of the feature was also reduced. This might lead to the assumption that basal melting is required to produce the height of basal crevasses observed near the calving front. However, previous modelling has shown that concentration of tensile stresses in the ice above a basal fracture can encourage further brittle fracture, providing a mechanism for basal crevasses to penetrate an increasing fraction of the ice thickness as they advect downstream (Bassis and Ma, 2015). In fact this may be a more likely mechanism for maintaining basal crevasse height, as other modelling work has shown that melt rates are likely to be highest on the walls of basal crevasses, while the peak of basal crevasses may be filled in by marine ice formation (Jordan et al., 2014; Khazendar and Jenkins, 2003). The relative importance of the two processes can only be investigated using a coupled model which could examine the local variations in basal melt rate, and the feedback from changes in ice geometry, which is beyond the current limits of the models presented here.

## 4 Conclusions

Our results represent the first application of a discrete element model, HiDEM, to an Antarctic ice shelf. We use the model to simulate fracture formation in two separate areas of the Totten IS. At the calving front, our results show that HiDEM reproduces fractures perpendicular to the calving front which contribute to the small, disintegration type calving events observed in satellite imagery. The model set-up at the calving front is not fully able to reproduce the fracture observed from satellite imagery, as it is missing many across-flow features which advect into the model domain from further upstream.

A second HiDEM simulation at the grounding line of the Totten IS demonstrates that this region is a likely source of basal fractures, particularly forming around pinning points. Simulations of crevasse evolution under homogeneous basal melting and longitudinal stretching demonstrate that narrow basal crevasses at the grounding line can plausibly evolve into the broader features observed near the calving front. However, coupled ocean/ice/fracture models would give a more thorough representation of the different processes involved in the evolution of basal crevasse shapes, and would provide a more robust comparison of the relative importance of ice shelf processes such as internal strain and basal melt.

Our results show that it is the interaction between the advected and locally produced fractures at the ice shelf calving front which controls the fracture pattern on the Totten IS, and thus controls the formation of icebergs. This presents a significant complication for accurately modelling calving at the Totten Glacier. The fracture pattern at the ice shelf front depends not only on the local lateral spreading which creates along-flow rifts. It is also strongly affected by across-flow features which are produced upstream, likely at the grounding line. This implies that the calving behaviour of the ice shelf is affected not only by stresses at the calving front, but also the cumulative ocean melt history acting on a section of ice shelf, and the conditions at the grounding line when the ice reached the shelf decades or centuries prior to calving. The combination of multiple different fracture sources is a significant challenge for calving models. One way to address this problem may be to combine modelling methods which allow local fractures to be accurately reproduced (e.g. HiDEM), in conjunction with traditional ice sheet modelling software which can track the history and advection of fractures over longer periods of time.

**Code availability**

The Elmer/Ice software is an open-source finite element for Ice Sheet, Glaciers and Ice Flow Modelling available at http://elmerice.elmerfem.org/ (Gagliardini et al., 2013). Other code used may be shared by the authors upon request.

**Data availability**

Ice geometry data from the ICECAP campaign are available online at NSIDC (Blankenship et al., 2015; Blankenship et al., 2012).

**Author contribution**

All authors contributed to experiment design. JA and TZ developed the model code and SC, JA and TZ performed simulations. SC led writing with contributions from all authors.

**Competing interests**

The authors declare that they have no conflict of interest.

# Acknowledgements

Computing resources were provided by CSC – IT Center for Science Ltd. and National Computational Infrastructure grant m68. Data collection was supported through funding from NSF grants PLR-0733025 and PLR-1143843, NASA grants NNG10HPO6C and NNX11AD33G (Operation Ice Bridge and the American Recovery and Reinvestment Act), the Australian

Government's Australian Antarctic Science Grant Program under projects AAS 3103, 4077 and 4346, the G. Unger Vetlesen Foundation, and the Jackson School of Geosciences. Our thanks to Dr. Jason Roberts, AAD for assistance in collating and processing data. This work was supported by the Australian Government's Business Cooperative Research Centres Programme through the Antarctic Climate and Ecosystems Cooperative Research Centre (ACE CRC) ) and the Australian Research Council Special Research Initiative for Antarctic Gateway Partnership project (Project ID SR140300001).

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

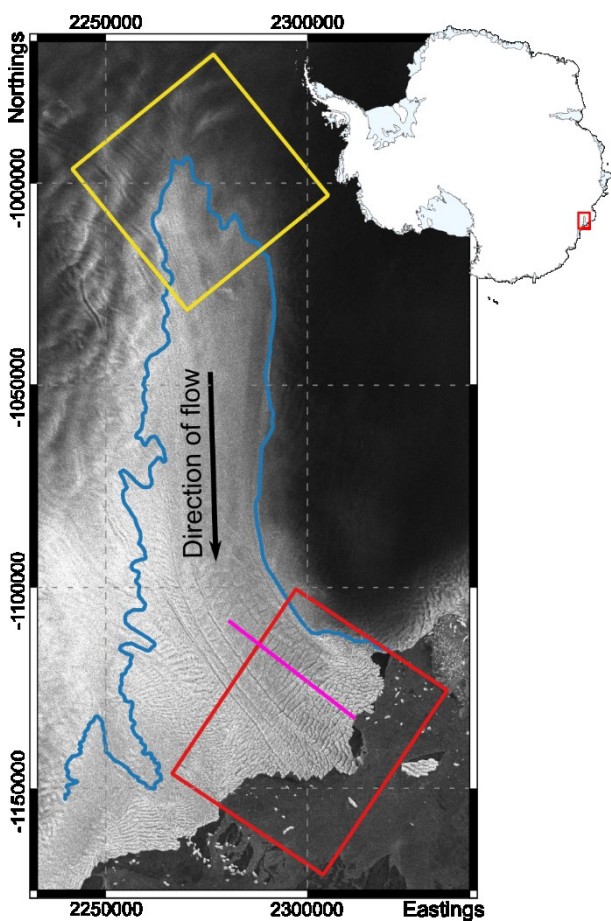

**Figure 1: Locations of model domains**. Blue line shows the grounding line as defined by ASAID (Bindschadler et al., 2011a). Red box shows extent of HiDEM calving front model. Yellow box shows extent of HiDEM grounding line model. Pink line shows flight path used to tune the KPZ model. Background image: Sentinel 1, December 2016.

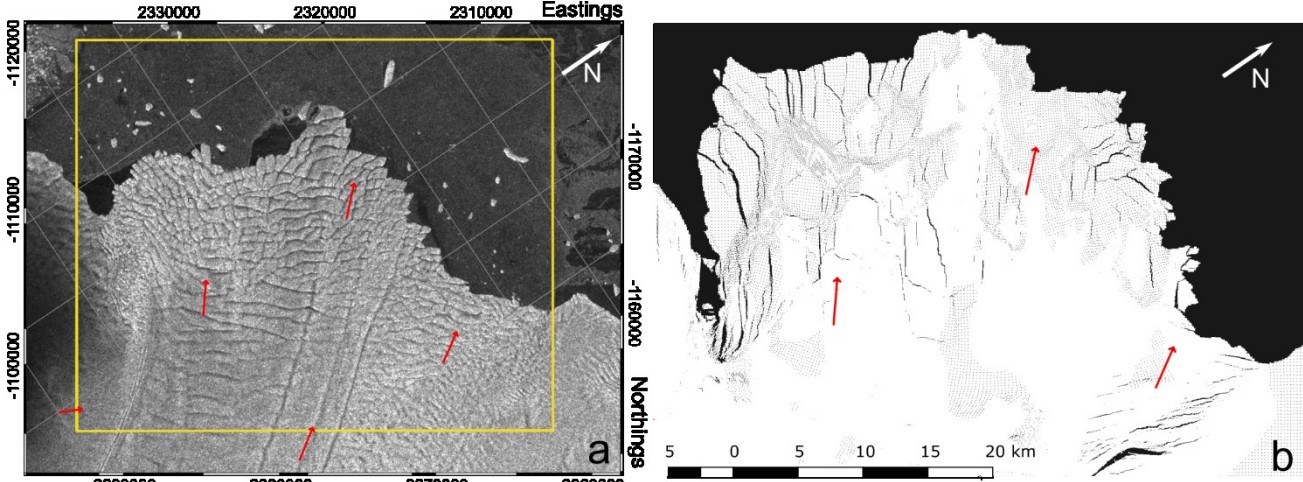

**Figure 2: Observed and modelled fractures. a.** Background image from Sentinel 1 (December 2016) shows the dense pattern of fractures at the calving front (yellow box indicates model domain) **b.** Modelled crevasses in HiDEM occur most densely near the calving front, and typically run roughly parallel to flow, or perpendicular to calving front. Red arrows show flow direction from MEaSUREs Antarctic velocity map (Mouginot et al., 2017a,b). In this projection thick ice appears white, thin ice appears grey and water appears black. The dark fractures on the shelf are full thickness rifts exposing the black water underneath.

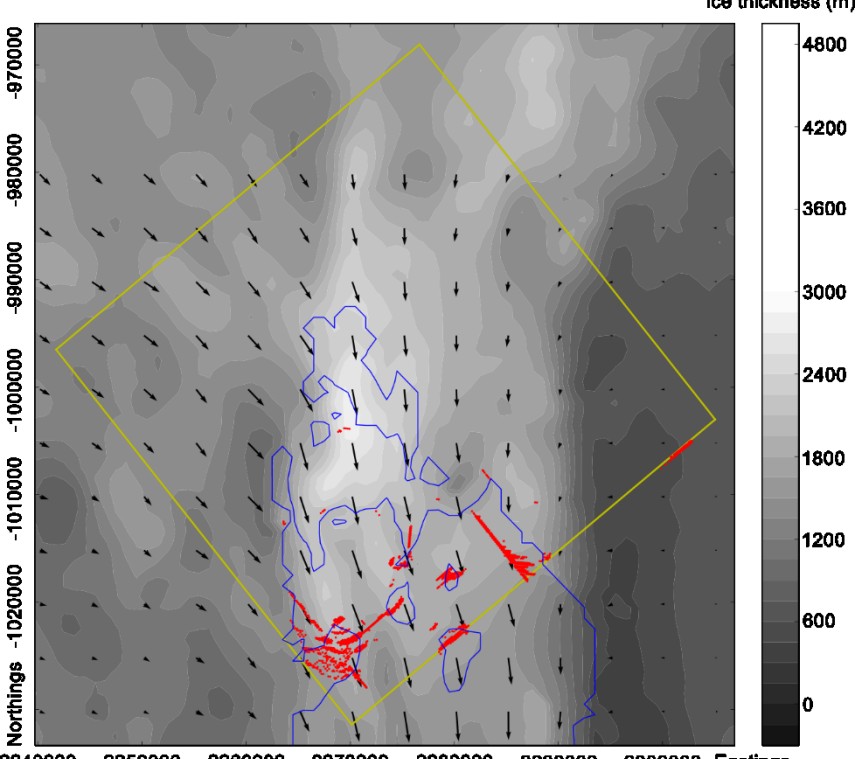

**Figure 3: Modelled fractures from HiDEM at the grounding zone.** Red dots show modelled crevasses. Blue line shows grounding line from HiDEM model geometry. Background greyscale shows ice thickness and arrows indicate ice flow direction from MEaSUREs (Mouginot et al., 2017a,b). Crevasses are largely clustered around pinning points where the ice re-grounds.

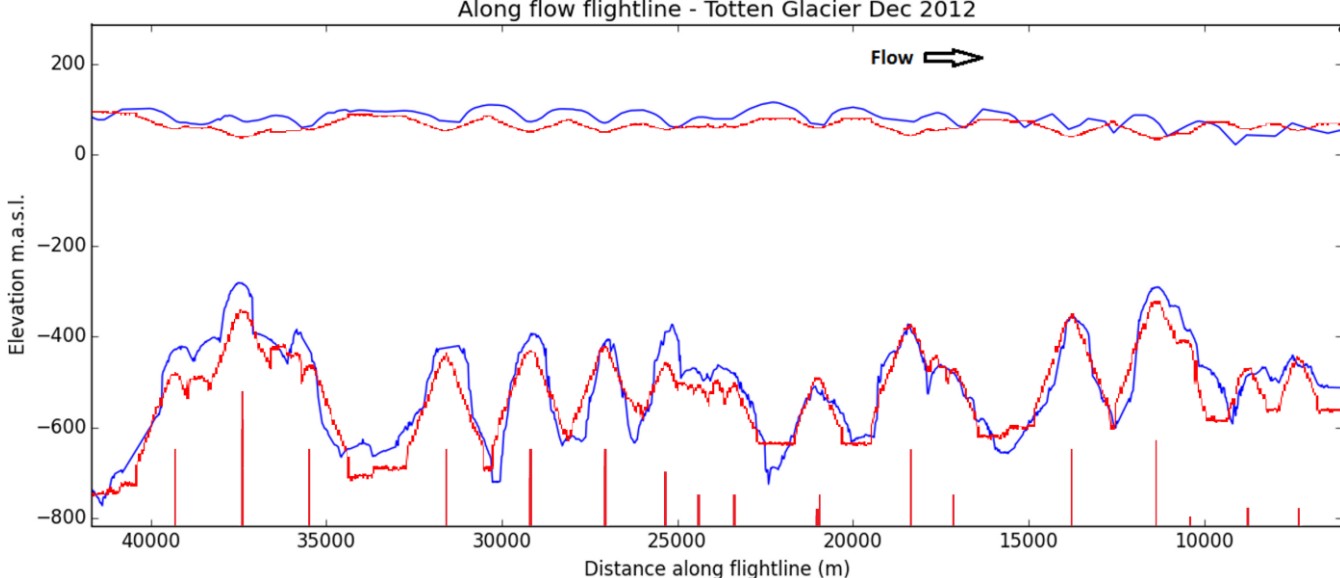

**Figure 4: Evolution of basal fracture geometry under ocean melt.** The KPZ equation is used to evolve an initial set of narrow fractures (vertical lines at the bottom of the figure) under constant basal melt. The computed surface and base after 100 years (red) can be made to match the measured geometry along a flight path (dark blue).

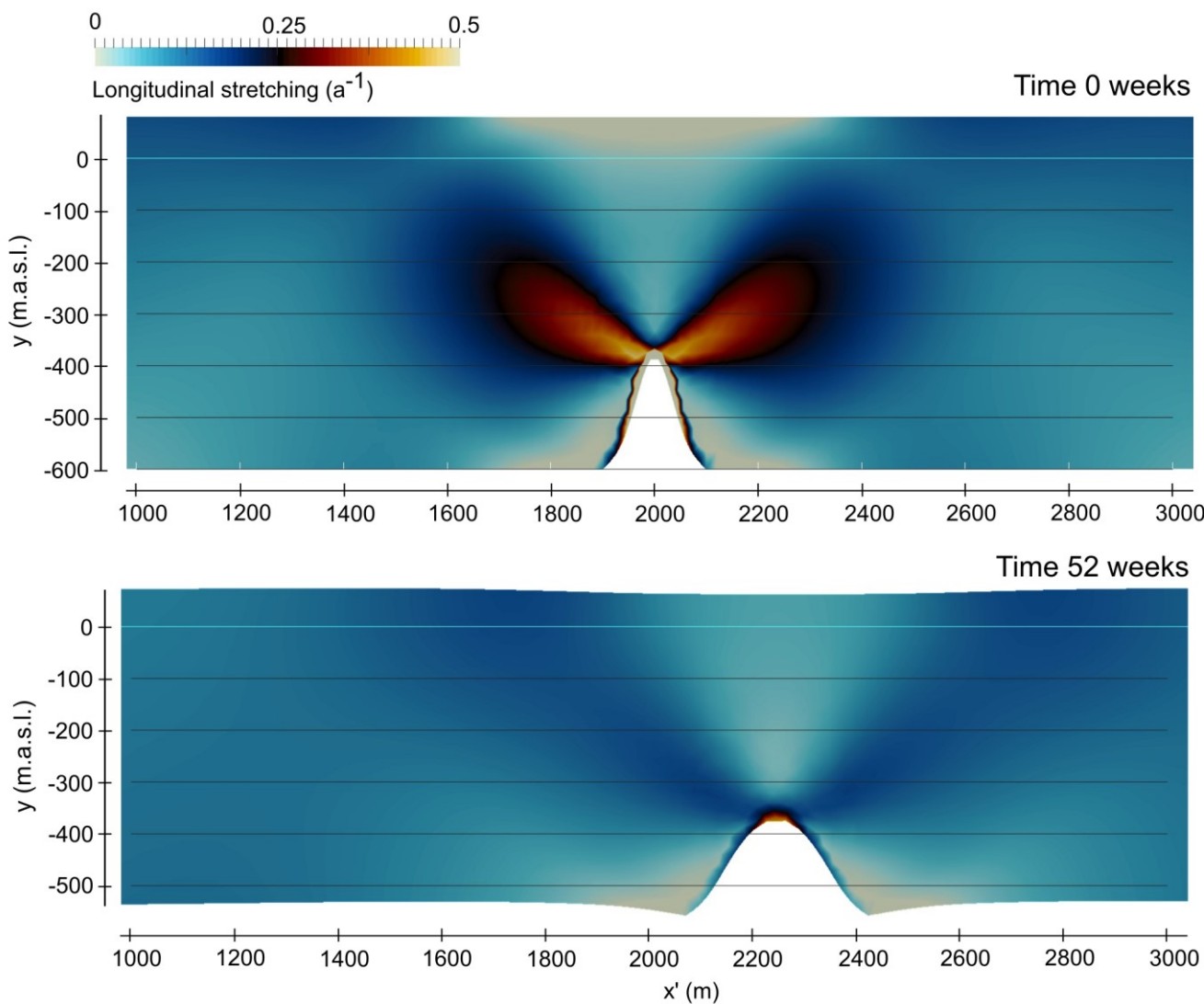

**Figure 5: Evolution of basal crevasse geometry in Elmer/Ice.** Top panel shows initial geometry of basal crevasse experiment. Bottom panel shows evolved geometry under longitudinal stretching after one year. The effect of longitudinal stretching on the shelf widens the basal crevasse.

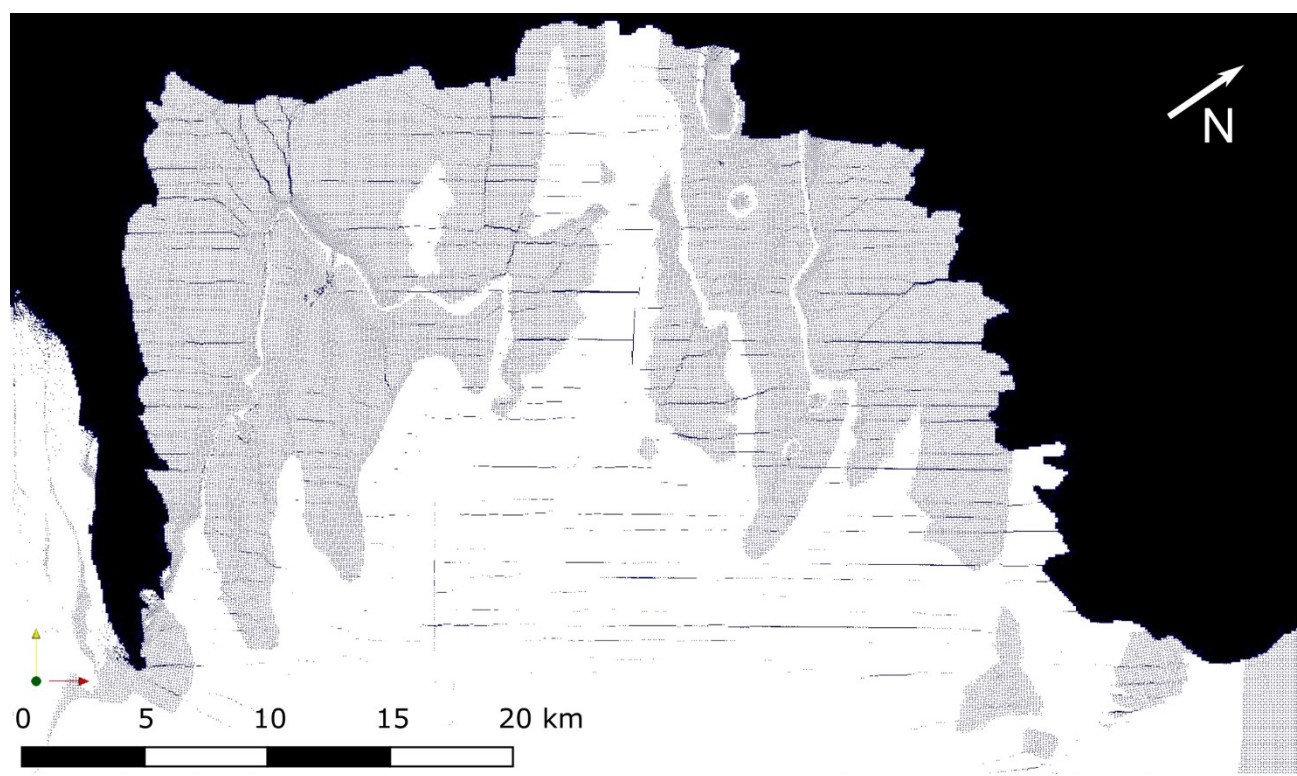

**Figure 6: Fracture pattern produced by HiDEM with artificial basal crevasses.** In this projection thick ice appears white, thin ice appears grey and fractures are visible as dark lines on the ice shelf. The introduced basal features are a source of additional across flow fracture, producing a checkerboard fracture pattern similar to observation.

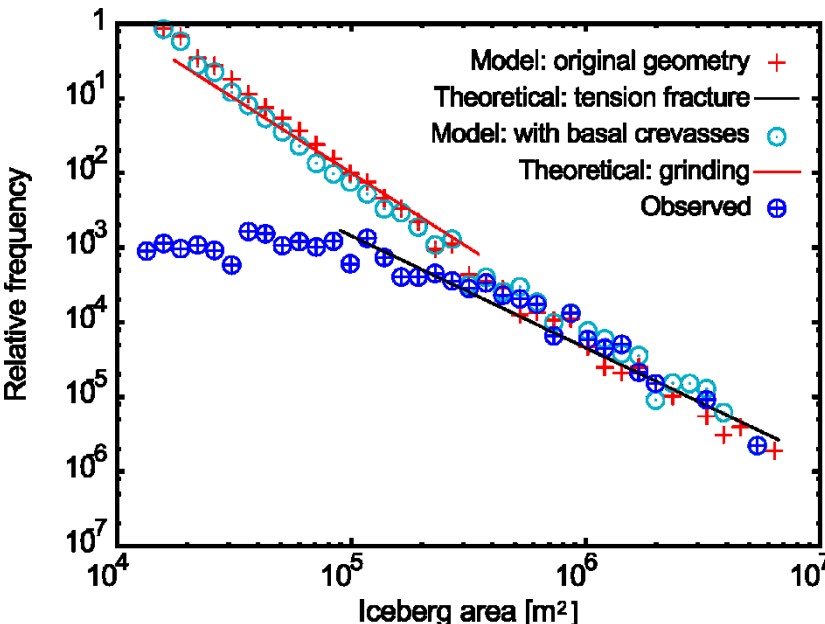

**Figure 7. Evaluation of iceberg size distributions.** Frequency distribution of simulated and observed iceberg sizes. Dark blue circles show observed iceberg sizes, red crosses show icebergs produced by the original model geometry, pale blue circles show icebergs produced by the modified geometry with additional basal crevasses. Overlaid are theoretical distributions for brittle fragmentation: area$^{-3/2}$ (black line) and grinding: area$^{-2}$ (red line).