# Peer review of "Modelled fracture and calving on the Totten Ice Shelf"

_The Cryosphere, 2018_

## Referee Comment (RC1) · J. H. Bondzio (Referee) · 22 Feb 2018

**Review on S. Cook et al., "Modelled fracture and calving on the Totten Ice Shelf" https://doi.org/10.5194/tc-2018-3**

J. H. Bondzio

February 21, 2018

**1 General Comments**

**1.1 Summary**

S. Cook et al. use three numerical models (HiDEM, Elmer\Ice, and a KPZ-model) to study the formation of crevasses on the Totten Ice Shelf (TIS). The TIS exhibits a complex crevasse pattern near the calving front of both along and across-flow crevasses. The authors find that the across-flow crevasses are not reproducible in the observed amount in a HiDEM simulation that includes the calving front region only. Priming the model through artificial introduction of across-flow crevasses into the model yields a good match of the observed iceberg size distribution, however. The authors argue that across-flow basal crevasses therefore are necessary to create the observed crevasse pattern, and have to be advected into the calving front region from upstream. In a second experiment using HiDEM, they find that basal across-flow crevasses form predominantly at regrounding points in the grounding zone. The authors argue consequently, using an implicit front tracking scheme (KPZ-model) and a continuum mechanics based model (Elmer\Ice), that initial narrow cracks created at the grounding zone are widened to observed widths as the ice shelf melts from beneath and spreads under its own weight.

**1.2 Result Novelty**

The model study presents novel and interesting results which should be published.

**1.3 Lack of a Sensitivity Study**

The authors use the numerical models to create a logical argumentation chain, which makes it plausible that basal across-flow crevasses are created at the grounding line, advected and widened toward the calving front, where they are

key for the calving process of this ice shelf. For my understanding, however, this model study lacks a solid sensitivity study of its model input parameters and parametrizations, i.o. to exclude other plausible explanations for the creation and role of across-flow crevasses. It is necessary to perform a sensitivity study w.r.t.:

- The yield stress parameter.
  You mention it is hard to reproduce the amount of across-flow crevasses, and therefore you argue that across-flow crevasses have to be created upstream at the grounding line. However, Figure 2b clearly shows the creation of across-flow crevasses upstream and even near the calving front in the model. For your argument, you need to rule out that this model "failure" to reproduce the amount of cracks is not due to e.g. the choice of a model parameter (e.g. the critical yield stress of 1 MPa (cf. p3, l24)), or other model simplifications (e.g. the lateral stress boundary condition, which excludes the driving stress). A lower critical yield stress might for example potentially result in more brittle failure of the ice upstream, increasing the number of across-flow crevasses created near the calving front. In my opinion, you need to test a range of yield stress parameters before you can infer from your results that the across-flow crevasses have to be created upstream.

- The basal friction law.
  You state you use a constant basal friction law. Using this law, a sharp stress gradient occurs at the grounding line, which is suitable to create basal crevasses. There is a lot of uncertainty regarding the basal conditions at the grounding line, however. Other basal sliding laws include the basal effective pressure (e.g. Budd et al., 1984), which would not create this basal stress gradient. You'd need to test different basal sliding laws before you can be certain to attribute the basal crevasses to the regrounding points.

- The choice of model domain extent.
  For the reason of limited ressources, you can not model the entire ice shelf. The inset of your model domain, however, shows that you do not include the eastern, grounded part of the TIS in your model domain. Certainly, this grounded area will have a significant impact on the momentum balance of the ice shelf, and possibly contribute to the creation of across-flow basal crevasses? From my understanding of the text, this region has also not been accounted for by imposing suitable boundary conditions. What are the implications of this?

**1.4 Model Setup Description**

For reproducibility it is important to include a comprehensive description of all the used models. All boundary conditions and used model input data sets need to be described. For the HiDEM model, please state the geometry and

ice velocity data sets used for setup. For the KPZ-model, please describe in detail the choice of parameters $\nu$, $\lambda$, and forcing $F$ (p5, l6). How do you vary $\eta$? How is the steady state (I'm assuming it is a steady state) shown in Figure 4 reached? A table listing all variables and model parameter values would be helpful.

**1.5    Paper Structure and Figure Clarity**

The paper loosely follows a Introduction-Methods&Data-Results-Discussion-Conclusions structure, without a separate discussion section, however. Parts of the discussion are found in the results and the conclusion section. This makes it at times hard to distinguish between the respective two. Moreover, it renders the conclusion part lengthy. I would suggest creating a separate discussion section, where the interesting model results are discussed at full length. The conclusion could then be shortened by summarizing the findings, and present an perspective for where future work should be performed. See specific comments below.

The clarity of some figures could be enhanced, cf. specific comments below.

**2    Specific Comments**

**2.1**

p3, l7: The introduction speaks about three models, HiDEM, Elmer\Ice and the KPZ equation. Which one is referred to here? Please be specific and describe for each model which data set is being used.

**2.2**

p3, l14: "Model performance": Please specify, which model performance is evaluated. That of HiDEM?

**2.3**

p3, l18: Title: The text speaks about discrete element models only. I suggest to change the section title to just that.

**2.4**

p4, l6: What are the reasons that you omitted the driving stress from the lateral stress boundary condition? Especially in the 2nd HiDEM model, the inclusion of the driving stress on grounded ice might make a significant difference for the stress regime.

**2.5**

p4, l17-19: Please state the type of basal friction law used, and add the unit of $c$.

**2.6**

p5, l2: With this KPZ-model, you assume uniformal melt along the surface of the basal crack. Are there observations that support this assumption? Some of the work you cited (Jordan et al., 2014) finds that the melting pattern inside a basal crevasse can be highly inhomogeneous, with melting at lower sections of the crevasse, and refreezing at higher ones. Therefore, it should be discussed whether the initial crack distribution $\eta$ is realistic, or whether perhaps deeper-penetrating cracks are possibly more realistic.

**2.7**

p6, l8 and elsewhere: The ice front geometry is complex. It is therefore difficult for me to distinguish between crevasses running parallel and perpendicular to the calving front: a crack can be perpendicular to one segment of the calving front and parallel to the other. Please specify how you define your terminology.

**2.8**

p6, l10-13: This is discussion material, please move to separate discussion section.

**2.9**

p6, l21-24: This is discussion material, please move to separate discussion section.

**2.10**

p7, l1-6: This is motivation, and should be in the introduction section.

**2.11**

p7, l7-12: This is model setup description, and should be in section 2.3.

**2.12**

p7, l14-17: This is discussion material, please move to separate discussion section.

**2.13**

p7, l20,21: Parts of the experiment description have been given earlier already in section 2.4.

**2.14**

p7, l22: Please add the duration it takes for this widening to take place. Figure 5 suggests 52 weeks?

**2.15**

p7, l23-p8, l4: Please move this discussion material to the discussion section.

**2.16**

p8, l7: Please add the spacing and length of the artificial across-flow basal crevasses that have been inserted. Ideally, this information would already be given in section 2.

**2.17**

p8, l5-13: Please move the model setup and discussion material to the respective sections.

**2.18**

p8, l10-13: Usage of many relative terms: "break apart easily", "fractures develop slowly". Please show the results for both cases so that the reader is able to compare.

**2.19**

p8, l13: Figure 7 does not show that the experiment without the insertion of basal crevasses produces the observed iceberg distribution. It only shows the case for where the basal crevasses have been inserted. I'd suggest to add the power spectrum of the earlier case to Figure 7.

**2.20**

p8, l19: Without a suitable sensitivity study, we cannot conclude that the across-flow crevasses have really to be advected from upstream.

**2.21**

p8, l24: You say that basal crevasses are created at the grounding line. With the HiDEM at hand, what can you say about the process that creates these basal crevasses? Is it - as I assume - the jump in basal shear stress?

**2.22**

p9, l9: Please list and discuss the implications of the "significant oversimplifications" at an earlier point.

**2.23**

p9, l13: "speed with which they form": This is new discussion material. Moreover, in the current manuscript, you do not show how fast the icebergs form in time, you only mention that they form "more quickly" once across-flow crevasses are introduced to the model (p8-l13). Videos in a supplementary material could be helpful to illustrate your point.

**2.24**

p9, l17: "which are likely produced at the grounding line": Again, you'd need to show first through a sensitivity study that the underproduction of across-flow crevasses in your calving-front-HiDEM setup is not due to the choice in model parameters.

**2.25**

Sect.4: Much of section 4 (p8, l20 to p9, l12-21) is discussion material, and would deserve a separate, earlier section. I suggest to then create the conclusions from a condensed version of the discussion and the outlook.

**2.26**

Figure 1: I suggest to add a scale ruler, a North arrow/grid lines, and an Antarctica location inset for orientation.

**2.27**

Figure 2: Only little information is gained by showing two times the same satellite image. The reader will still get a feel for the crevassing pattern if image a was to be left out. Even more so, as the remaining image would be printed larger, and the crevassing pattern would be visible in more detail.

**2.28**

Figure 3: The main information of this figure should be the basal crevasses, which occupy only the bottom part of the plot. The most visible information is the colorful ice thickness, which stems from a data set not presented here. I'd therefore suggest to use a less dominating color map for the ice thickness (e.g. a gray scale). The crevasses (black) and the grounding line (dark gray) are hard to distinguish. I'd suggest using a brighter color for the crevasses.

**2.29**

Figure 4: Just a comment: The blue color of the vertical lines at the bottom associates them with the observed blue profile. I suggest changing their color to red, i.o. to associate them with the modeled profile. Please elaborate: does this figure show a steady state geometry?

**2.30**

Figure 5: The clarity of the figure could be enhanced by a) cropping the x and y axis so that the floating slab of ice is in the respective center of the subplot, b) dropping the 6-digit precision of the time which is superfluous, c) using a uniform format for the ticklabels of the colorbar, and d) including a y-axis.

**2.31**

Figure 6: This figure was confusing to me at first due to its grainy appearance. The meaning of the discontinuous horizontal black lines is not clear. The broken bonds between the ice bergs are hard to distinguish from only slightly damaged or undamaged ice. The Cartesian three-arrow orientation guide has no labels and could be left out. Additionally, x and y-labels, a scaleruler, gridlines and/or a northarrow would be helpful for orientation for the reader. Why are some of the areas up to the calving front white (and I assume not damaged?). I think it would be useful - for comparison of the crevasse pattern - to include the same figure next to it for the case without the added basal across-flow crevasses. Why not use the same layout at Figure 2b?

**2.32**

Figure 7: The plot includes a fit to the iceberg size distribution. Are you trying to fit the observations or the model results? Are you able say anything about the exponent chosen for the fit, or conclude anything from it? Please discuss.

**3 Minor Corrections**

**3.1**

p5, l9: This sentence is incomplete. Add "it"?

**3.2**

p6, l4: 2000 km$^2$: For consistency, please use side lengths like for the second model domain.

**3.3**

p9, l17: "This implies..": The sentence does not flow. Please rephrase.

**References**

Budd, W., D. Jenssen, and I. Smith
   1984. A three-dimensional time-dependent model of the West Antarctic ice-
   sheet. *Ann. Glaciol.*, 5:29–36.

Jordan, J. R., P. R. Holland, A. Jenkins, M. D. Piggott, and S. Kimura
   2014. Modeling ice-ocean interaction in ice-shelf crevasses. *J. Geophys. Res.*,
   119(2):995–1008.

---

## Referee Comment (RC2) · J. Bassis (Referee) · 26 Feb 2018

Review: Cook et al.,

**General Appreciation.**

This manuscript describes a suite of models used to describe the fracture patterns observed around Totten Ice Shelf, East Antarctica. The authors apply a discrete element models (HiDEM) a full Stokes model (Elmer/Ice) and a model meant to describe basal melt (the KPZ equation) to describe the formation and evolution of basal crevasses. The authors use the suite of models to infer that basal crevasses that initiate near the grounding zone advect to the calving front and are responsible for the checker board pattern of fractures observed near the calving front of the ice shelf. That basal crevasses that initiate upstream play an important role in calving closer to the calving front is not that surprising and is consistent with several prior studies. However, this

is an interesting hypothesis—especially in light of the argument that the channel geometry is related to submarine melt—because it provides a direct connection between fractures/calving and ocean forcing and this study provides the first direct numerical simulation of some of the fundamental processes. Hence, the study has the potential to be improve our understanding of a set of processes that remain poorly understood.

Organizationally, I found the manuscript a bit confusing to follow. I think it would help readers significantly if the authors presented their hypotheses or questions earlier on in the manuscript. On my first reading, it seemed as though the authors were merely throwing a couple of unrelated models at a problem without any underlying reason or rational. It wasn't until near the end of the manuscript that I realized that the authors were testing the hypothesis that crevasses near the grounding line control the fracture pattern near the calving front. That the authors need the basal crevasses is interesting and should be emphasized to readers earlier on. Here, I think the authors could significantly improve the impact of the manuscript by adding section subtitles and sketching out the logical flow. In other words, it would be helpful to see that the authors are trying to explain the fracture distribution near the calving front of the ice shelf. This could either be presented as a fundamental question early on (i.e., basal crevasses are needed to explain the fracture pattern), or complexity could be introduced into the suite of models expressly to match this fracture pattern (i.e., the HiDEM simulation without basal crevasses could not represent the fracture pattern). Without this additional text, some of the models appear unmotivated. I do also have some questions about the models and assumptions, described in more detail below.

**Questions about models.**

Overall, I think that the hierarchy of models is interesting. I do have a few questions about the models. For example:

-How many particles are used in the HiDEm model runs. What is the vertical number of particles? More details on the numerical setup would be helpful to readers.

-How are the basal crevasses/channels used in the last simulation specified in HiDEM? Are these based on observations? Providing this data is critical.

-What ice temperature is used in the ELMER/ICE simulations? This should be important in determining the rate of crevasse widening. I'm also curious about the effect of the strain rate on widening. The strain rate is sufficient to allow crevasses to widen, but I would expect crevasses to actually close if the strain rate is smaller. This would be interesting to check.

-Similarly, I would have liked to see some more quantitative comparisons between observations and data if feasible. For instance, how wide are the fractures predicted by HiDEm and do they penetrate the entire ice thickness? It is difficult to detect if fractures penetrate the entire ice thickness using surface observations, but one could estimate the width of fractures (based on pixels) and compare this to the predicted width of fractures from HiDEM. In fact, it would be really nice to have a statical comparison between length, width and orientation of fractures predicted by HiDEM and those observed in the images. This would significantly bolster the authors claims and avoid the need to rely on qualitative inspection of figures to determine how well the model is performing. This is important for a range of reasons, but it is important to recognize that the iceberg size distribution is modified by interaction with the ocean (bergs melt, erode, tip over, etc.) and hence the comparison between berg sizes is not necessarily a one-to-one comparison with bergs that detach. Fracture distributions should be more straightforward to map and compare.

A few other quantitative questions I had about data included: How many total icebergs were detached in 2009, 2010 and 2011 respectively and how many icebergs in each size class were detected? What time of year are these images? Is there a bias associated with the time of year? Furthermore, it is possible that the distribution of icebergs/calving rate is not stationary stochastic. If this is the case, then the distribution of icebergs would not be stationary stochastic, as the authors assume.

**KPZ equation** The use of the KPZ equation in this context is interesting and novel. I have played around with it a few times for various things. My understanding of the usage of the KPZ equation here is the $\nu$, $\lambda$ and $F$ (which includes two terms to allow for a linearly varying profile) are all adjusted to match observations. This means that the authors have 4 free parameters to adjust. From looking at Figure 4, it also looks like the position, width and initial depth of the crevasses are all also determined as part of the solution. There is an old saying that given three parameters you can fit an elephant. Given 5, you can make its trunk wiggle. Although, I find this a bit extreme, I do think it is worth discussion the robustness of the results in the face of 7 unconstrained parameters. This should start by listing the numerical values of parameters used. Are results sensitive to order of magnitude changes in parameters? Factors of two changes? I also have questions about the fact that, as far as I can tell the melt rate ($F$) is not related to ocean forcing at all. I guess this is just an empirical constant, but the KPZ model is of limited utility if we cannot relate it to ocean forcing. At the very least, we have observations of the melt rate and hence, can constrain the average value of the melt forcing term. Similarly, the time at which the simulation result is obtained is not given. How long does it take for these channels to evolve and how different do the channels look at different instances of time. This is crucial because the KPZ equation does not necessarily admit a steady-state—similarity solutions exist that allow for continuous variation. If the solution is steady-state, then this should be declared along with how long it takes the simulation to arrive at steady-state.

As I mentioned, I think the use of the KPZ equation here is interesting and novel. However, it appears as though the KPZ equation as listed in Equation (4) has an error. The actual equation should read:

$$\frac{\partial h}{\partial t} + u \cdot \nabla h = \nu \nabla^2 h + \frac{\lambda}{2} \left[\nabla h\right]^2 + F + \eta. \tag{1}$$

where $u$ is the velocity along the interface. There is no physical reason to omit the advection term. Typically, the KPZ equation is used for materials where the interface

does not have a background velocity (as in the refs cited) and hence, one can replace the material derivative with the partial derivative with respect to time, as the authors seem to have done. In this case, however, the interface is moving with the background velocity of the ice and there is a gradient in the velocity. If the "fractures" are indeed enlarged by a constant melt rate, then including the advection term should allow the authors to reproduce the spatial pattern of fractures using a constant melt rate $F$ instead of a linearly varying melt rate. Now, because velocity is not Galilean invariant one need only specify the difference in velocity across the domain. One could do this in a consistent manner with the Elmer/Ice model by using the same strain rate across the domain.

There are other, subtle, issues associated with the use of the KPZ equation in this context. One prominent assumption of the KPZ equation is that the erosion of the interface does not depend on the absolute elevation of the interface. This is the reason why only scalar powers of $\nabla h$ appear in the expansion. In the case of submarine melt, however, this assumption is violated because of the pressure dependence of the melting point. Erosion of ice should occur more rapidly deep beneath the surface of the ocean and erosion (or even marine ice deposition) should occur closer to the ocean surface. As a consequence, I'm not even sure that the *sign* of the $[\nabla h]^2$ term is right. Expanding on this point, the KPZ equation is useful for describing interface evolution because it represents a large universality class of processes that, irrespective to process details, ultimately behave like the KPZ equation. The universality class, however, is not infinitely large. Here, I think the authors should attempt to establish whether submarine melt actually resides in the broader universality class exemplified by the KPZ equation. Failing this, I would really like more detail on \*\*why\*\* the authors think the KPZ equation is an adequate description of the submarine melting process and the role of the pressure dependance of the melt point in this argument.

Finally, I was confused about the role of the Elmer/Ice and KPZ simulations of basal channel evolution. Why not incorporate the KPZ equation into the ELMER/ICE simulations so that the authors can self-consistently simulate the strain induced widening and erosion? If this isn't feasible, then what are we supposed to learn from the two simulations? As far as I can tell, the Elmer/Ice simulation tells us that crevasses widen and become shallower over time, but the Elmer/Ice simulations omit ocean forcing. In contrast, the KPZ simulations tell us that crevasses become wider and deeper over time, but this simulation omits any ice dynamics (included advection!). Do these effects cancel out? Does one dominate? Do we need both or can we omit one of them? I do wonder if the widening by ice dynamics alone is sufficient to explain the width of observed basal crevasses. If it is not sufficient, then this argues that erosion by submarine melt is necessary.

Detailed comments:

Page 3: "We also use a continuum ice flow model and a stochastic equation describing fracture development" This is unclear. It seems as though the KPZ equation is used to model the evolution of the interface in response to melt. The KPZ equation is indeed stochastic. However, the authors omit the stochastic terms, at which point version of the KPZ that they use is entirely deterministic . . . Also, the authors claim $\eta$ is a stochastic term, but then they set it equal to the initial width/length of basal crevasses. This seems to confuse initial conditions with the stochastic forcing. Typically, the stochastic term in the KPZ emerges as a white noise Gaussian process, which is quite different from the assumptions made here. It might be less confusing to separate this into an initial condition and noise term with the noise term set to zero.

This is a minor point given the fact that the stochastic term is set to zero, but stochastic partial differential equations fall into two categories the so-called Ito and Stratonovich interpretations. These two interpretations are subtly different in important ways, but necessary to place stochastic differential equations on firm mathematical footing. What interpretation is assumed here?

Page 4: How big of a difference does it make to results if a different percentage of

bonds is broken? Say 0.1% or 10%?

Page 4 last sentence after equation 3. The constant "c" is apparently dimensionless (at least no units are given) and thus friction does not have the correct units. Either "c" should have units or units of friction needs to be more carefully explained (have the equations themselves been non-dimensionalized)?

Figure 2: Am I missing something? Fractures look like the run both parallel and perpendicular to the front?

Figure 3 is hard to decipher. Is it possible to show Figure 3 in the same style of as Figure 2 so that we can see where the crevasses are located on an image?

Figure 6 is very pixelated (maybe a conversion issue?) and hard to interpret.

Figure 7. It doesn't look like the simulation obeys a -3/2 scaling law. According to the red curve, the computation predicts more icebergs at low iceberg area and not enough at large iceberg area. The observations are really only power-law over about an order of magnitude. Fitting power-laws to such a limited data range is fraught with peril. This maybe an example where it is better to avoid a logarithmic plot AND to include error bars in the observations. How many icebergs are there with sizes close to $10^6$ m$^2$? Is this actually statistically significant?

[Figure]

---

## Author Comment (AC1) · 9 May 2018

We thank both reviewers for their thorough and thoughtful comments. We have made significant changes to the structure of the paper in response. Replies to the reviewers' comments are shown in blue below. The revisions to the manuscript have resulted in changes to line numbers and equivalent new line numbers are provided with each response.

**Reviewer 1: J. H. Bondzio**

**1 General Comments**
**1.1 Summary**
S. Cook et al. use three numerical models (HiDEM, Elmer/Ice, and a KPZ-model) to study the formation of crevasses on the Totten Ice Shelf (TIS). The TIS exhibits a complex crevasse pattern near the calving front of both along and across-flow crevasses. The authors find that the across-flow crevasses are not reproducible in the observed amount in a HiDEM simulation that includes the calving front region only. Priming the model through artificial introduction of across-flow crevasses into the model yields a good match of the observed iceberg size distribution, however. The authors argue that across-flow basal crevasses therefore are necessary to create the observed crevasse pattern, and have to be advected into the calving front region from upstream. In a second experiment using HiDEM, they find that basal across-flow crevasses form predominantly at regrounding points in the grounding zone. The authors argue consequently, using an implicit front tracking scheme (KPZ-model) and a continuum mechanics based model (Elmer/Ice), that initial narrow cracks created at the grounding zone are widened to observed widths as the ice shelf melts from beneath and spreads under its own weight.

**1.2 Result Novelty**
The model study presents novel and interesting results which should be published.

**1.3 Lack of a Sensitivity Study**
The authors use the numerical models to create a logical argumentation chain, which makes it plausible that basal across-flow crevasses are created at the grounding line, advected and widened toward the calving front, where they are key for the calving process of this ice shelf. For my understanding, however, this model study lacks a solid sensitivity study of its model input parameters and parametrizations, i.o. to exclude other plausible explanations for the creation and role of across-flow crevasses. It is necessary to perform a sensitivity study w.r.t.:

- The yield stress parameter.
  You mention it is hard to reproduce the amount of across-flow crevasses, and therefore you argue that across-flow crevasses have to be created upstream at the grounding line. However, Figure 2b clearly shows the creation of across-flow crevasses upstream and even near the calving front in the model. For your argument, you need to rule out that this model "failure" to reproduce the amount of cracks is not due to e.g. the choice of a model parameter (e.g. the critical yield stress of 1 MPa (cf. p3, l24)), or other model simplifications (e.g. the lateral stress boundary condition, which excludes the driving stress). A lower critical yield stress might for example potentially result in more brittle failure of the ice upstream, increasing the number of across-flow crevasses created near the calving front. In my opinion, you need to test a range of yield stress parameters

before you can infer from your results that the across-flow crevasses have to be created upstream.

We understand the reviewer's desire to see more thorough sensitivity testing. However, each model run requires approximately 100,000 CPU hours to complete, making a suite of sensitivity tests unfeasible for this model geometry. The model has been tested extensively in other investigations to find best-fit parameters for glacier ice (see references below). For the model to produce plausible results there is not much room to vary the yield stress. If the yield stress is too low the ice shelf simply disintegrates.

Some variations can be allowed and the reviewer is correct that more crevasses would appear as yield stress is lowered. However, the effect is to increase the frequency of the cracks appearing but without changing the underlying orientations of the fractures. We would not be able to create the observed across-flow fractures in this way.

A discussion of this issue has now been added to the text (p4 l22 – p5 l1).

Åström, J. A., Riikilä, T. I., Tallinen, T., Zwinger, T., Benn, D., Moore, J. C. and Timonen, J.: A particle based simulation model for glacier dynamics, Cryosphere, 7(5), 1591–1602, doi:10.5194/tc-7-1591-2013, 2013.

Åström, J. A., Vallot, D., Schäfer, M., Welty, E. Z., O'Neel, S., Bartholomaus, T. C., Liu, Y., Riikilä, T. I., Zwinger, T., Timonen, J. and Moore, J. C.: Termini of calving glaciers as self-organized critical systems, Nat. Geosci., 7(12), 874–878, doi:10.1038/ngeo2290, 2014.

Benn, D. I., Åström, J. A., Zwinger, T., Todd, J., Nick, F. M., Cook, S., Hulton, N. R. J. and Luckman, A.: Melt-under-cutting and buoyancy-driven calving from tidewater glaciers: new insights from discrete element and continuum model simulations, J. Glaciol., 1–12, doi:10.1017/jog.2017.41, 2017.

Riikilä, T. I., Tallinen, T., Åström, J. and Timonen, J.: A discrete-element model for viscoelastic deformation and fracture of glacial ice, Comput. Phys. Commun., (APRIL), 13–22, doi:10.1016/j.cpc.2015.04.009, 2014.

Vallot, D., Åström, J., Zwinger, T., Pettersson, R., Everett, A., Benn, D. I., Luckman, A., Pelt, W. J. J. Van and Nick, F.: Effects of undercutting and sliding on calving : a coupled approach applied to Kronebreen , Svalbard, Cryosph. Discuss., doi:doi.org/10.5194/tc-2017-166, 2017.

- The basal friction law.
  You state you use a constant basal friction law. Using this law, a sharp stress gradient occurs at the grounding line, which is suitable to create basal crevasses. There is a lot of uncertainty regarding the basal conditions at the grounding line, however. Other basal sliding laws include the basal effective pressure (e.g. Budd et al., 1984), which would not create this basal stress gradient. You'd need to test different basal sliding laws before you can be certain to attribute the basal crevasses to the regrounding points.

It is not currently possible to include complex sliding laws such as that of Budd et al. in the HiDEM model. Instead, the constant basal drag coefficient we use is very low (we use a value of $10^7$ kg m$^{-2}$ s$^{-1}$, as compared to typical flow model inversion values ~$10^{11}$ kg m$^{-2}$ s$^{-1}$ e.g. Morlighem et al., 2013; Arthern et al., 2015). This choice of basal friction is successful in reducing the effects of a steep gradient in basal drag at the grounding line, as demonstrated by the fact that crevasses do not appear at a grounding line. Instead, basal crevasses are concentrated at re-grounding islands, where the glacier experiences compression and an upwards push as the ice hits the island, while on top and lee side of the island there is

tension. These will create fractures near the base independent of the basal friction parameter chosen. This is now discussed in the text (p10 l24 – p11 l5).

Morlighem, M., et al. "Inversion of basal friction in Antarctica using exact and incomplete adjoints of a higher-order model." Journal of Geophysical Research: Earth Surface 118.3 (2013): 1746-1753.

Arthern, Robert J., Richard CA Hindmarsh, and C. Rosie Williams. "Flow speed within the Antarctic ice sheet and its controls inferred from satellite observations." Journal of Geophysical Research: Earth Surface 120.7 (2015): 1171-1188.

- The choice of model domain extent.
  For the reason of limited resources, you cannot model the entire ice shelf. The inset of your model domain, however, shows that you do not include the eastern, grounded part of the TIS in your model domain. Certainly, this grounded area will have a significant impact on the momentum balance of the ice shelf, and possibly contribute to the creation of across-flow basal crevasses? From my understanding of the text, this region has also not been accounted for by imposing suitable boundary conditions. What are the implications of this?

In the data set we use there is a larger grounded section of the shelf on the west side and a smaller one on the east side, a little upstream from the terminus (as shown in the figure below with floating ice in red and grounded ice in green). We have used an isostatic condition on the boundaries of the domain (Equation 2, p5) which is a simplification of the driving stress, as it cannot account for longitudinal stresses, which can be an important factor in the stress balance around the grounding line and transition zone. The limited domain is therefore a weakness of our model. We do not have the resources to rerun results for a larger region, but have now included this limitation in the discussion section (p10 l17-21).

[Figure]

**1.4 Model Setup Description**
For reproducibility it is important to include a comprehensive description of all the used models. All boundary conditions and used model input data sets need to be described. For the HiDEM model, please state the geometry and ice velocity data sets used for setup. For

the KPZ-model, please describe in detail the choice of parameters $v$, $\lambda$, and forcing $F$ (p5, l6). How do you vary η? How is the steady state (I'm assuming it is a steady state) shown in Figure 4 reached? A table listing all variables and model parameter values would be helpful.

Some of this information was missing in the text and has now been added as follows:

- HiDEM geometry: The geometry of the HiDEM model is taken from surface elevation and ice thickness data collected by the ICECAP program. These are interpolated using TELVIS (Thickness Estimation by a Lagrangian Validated Interpolation Scheme) see Sect. 2.1.
- Ice velocity: The ice velocity is not used as model input. Instead, the boundary conditions are set using isostatic/hydrostatic pressure (Equation 2, p5).
- KPZ parameters: The parameters $v$=1 a$^{-1}$, $\lambda$=1 ma$^{-1}$ and $F = 1 x 10^{-3} x'$ma$^{-1}$ are now included in the text (p7 l9&11).
- Choice of η: η is used in this experiment to initialise the basal fractures in the model, after which time η(t>0) = 0 (p7 l8-10). The initial fractures are chosen to most closely match the observed geometry, and can be seen in Figure 4.
- The KPZ model output is not shown in steady state. Instead, we stopped the experiment after 100 years, which is roughly the transit time from grounding line to calving front for the flight path shown in Figure 1. This is now explicitly stated in the text (p9 l10).

**1.5 Paper Structure and Figure Clarity**

The paper loosely follows a Introduction-Methods&Data-Results-Discussion-Conclusions structure, without a separate discussion section, however. Parts of the discussion are found in the results and the conclusion section. This makes it at times hard to distinguish between the respective two. Moreover, it renders the conclusion part lengthy. I would suggest creating a separate discussion section, where the interesting model results are discussed at full length. The conclusion could then be shortened by summarizing the findings, and present a perspective for where future work should be performed. See specific comments below. The clarity of some figures could be enhanced, cf. specific comments below.

The structure of the paper has been altered as suggested, to include an explicit discussion section and to improve the clarity of discussion of different models.

**2 Specific Comments**

**2.1**: p3, l7: The introduction speaks about three models, HiDEM, Elmer/Ice and the KPZ equation. Which one is referred to here? Please be specific and describe for each model which data set is being used.

This has now been clarified (p4 l2) and each model description now states whether it uses the observed or an idealised geometry (p6 l1, p7 l8, p7 l25).

**2.2**: p3, l14: "Model performance": Please specify, which model performance is evaluated. That of HiDEM?

This has now been clarified (p4 l11).

**2.3**: p3, l18: Title: The text speaks about discrete element models only. I suggest to change the section title to just that.

The section title has been changed as suggested (p4 l17).

**2.4**: p4, l6: What are the reasons that you omitted the driving stress from the lateral stress boundary condition? Especially in the 2nd HiDEM model, the inclusion of the driving stress on grounded ice might make a significant difference for the stress regime.

The lateral boundaries of the model are forced using isostatic pressure from the surrounding ice (Equation 2). This is a simplification of the driving stress, as it cannot account for longitudinal stresses, which can be an important factor in the stress balance around the grounding line and transition zone (e.g. Pattyn et al. 2006, Schoof 2007). Were these stresses to be included it is possible that the fracture pattern would change, and this is now acknowledged within the text (p5 l17-18 & p10 l17-21).

Ref: Pattyn, Frank, et al. "Role of transition zones in marine ice sheet dynamics." Journal of Geophysical Research: Earth Surface 111.F2 (2006).

**2.5**: p4, l17-19: Please state the type of basal friction law used, and add the unit of c.

This has been altered in the text (p5 l18-22).

**2.6**: p5, l2: With this KPZ-model, you assume uniform melt along the surface of the basal crack. Are there observations that support this assumption? Some of the work you cited (Jordan et al., 2014) finds that the melting pattern inside a basal crevasse can be highly inhomogeneous, with melting at lower sections of the crevasse, and refreezing at higher ones. Therefore, it should be discussed whether the initial crack distribution $\eta$ is realistic, or whether perhaps deeper-penetrating cracks are possibly more realistic.

The melting assumed in the KPZ is highly simplistic, and we have now expanded the text to discuss in more detail the implications of this for the development in shape of the basal crevasses (p7 l13-18 & p12 l7-10).

**2.7**: p6, l8 and elsewhere: The ice front geometry is complex. It is therefore difficult for me to distinguish between crevasses running parallel and perpendicular to the calving front: a crack can be perpendicular to one segment of the calving front and parallel to the other. Please specify how you define your terminology.

We have updated Figure 2 to show results from a later timestep, which should make it easier for the reader to observe the overall trend of fracture orientation. Not all fractures are perpendicular to the calving front, and the text has been amended to make this clearer (p8 l11-15). What can easily be observed in the figure is the lack of across-flow fractures, particularly upstream of the calving front. This is also analysed in the new supplementary material using the observed fracture orientations.

**2.8**: p6, l10-13: This is discussion material, please move to separate discussion section.

This now appears in a new Discussion section (p10 l22-23).

**2.9**: p6, l21-24: This is discussion material, please move to separate discussion section.

This now appears in a new Discussion section (p11 l6-11).

**2.10**: p7, l1-6: This is motivation, and should be in the introduction section.

This has been moved to the Introduction (p2 l25 – p3 l2).

**2.11**: p7, l7-12: This is model setup description, and should be in section 2.3.

This has been moved to Sect 2.3 (p7 l7-12).

**2.12**: p7, l14-17: This is discussion material, please move to separate discussion section.

This now appears in a new Discussion section (p12 l3-10).

**2.13**: p7, l20,21: Parts of the experiment description have been given earlier already in section 2.4.

Areas of repetition have been removed.

**2.14**: p7, l22: Please add the duration it takes for this widening to take place. Figure 5 suggests 52 weeks?

This has been added to the text (p9 l16).

**2.15**: p7, l23-p8, l4: Please move this discussion material to the discussion section.

This now appears in a new Discussion section (p11 l13-16).

**2.16**: p8, l7: Please add the spacing and length of the artificial across-flow basal crevasses that have been inserted. Ideally, this information would already be given in section 2.

We used trigonometric functions to mimic depth, width and spacing of crevasses in Fig.4. The equation is now given in the text (Equation 4) and has been moved to Sect. 2.2 (model setup).

**2.17**: p8, l5-13: Please move the model setup and discussion material to the respective sections.

This section has been rewritten, and relevant text has been moved to Sect. 2.2 and the Discussion.

**2.18**: p8, l10-13: Usage of many relative terms: "break apart easily", "fractures develop slowly". Please show the results for both cases so that the reader is able to compare.

We have rewritten the text to make our statements more robust, and limited them to the data shown in Figure 7 (p9 l20-23).

**2.19**: p8, l13: Figure 7 does not show that the experiment without the insertion of basal crevasses produces the observed iceberg distribution. It only shows the case for where the basal crevasses have been inserted. I'd suggest to add the power spectrum of the earlier case to Figure 7.

The figure has been amended as suggested.

**2.20**: p8, l19: Without a suitable sensitivity study, we cannot conclude that the across-flow crevasses have really to be advected from upstream.

See above discussion.

**2.21**: p8, l24: You say that basal crevasses are created at the grounding line. With the HiDEM at hand, what can you say about the process that creates these basal crevasses? Is it - as I assume - the jump in basal shear stress?

The basal crevasses are primarily produced not at the initial grounding line, but at re-grounding points downstream. The fractures are caused by the upward gradient in the bedrock, which causes compression of the ice on the upstream face of the island, and conversely extension on the top and lees-side of the island. It is this extension that leads to the across-flow basal crevasses. The along-flow crevasses are caused by shear.

This is now discussed in more detail in the text (p10 l24 – p11 l5).

**2.22**: p9, l9: Please list and discuss the implications of the "significant oversimplifications" at an earlier point.

These are now discussed earlier in the text (p7 l13-18).

**2.23**: p9, l13: "speed with which they form": This is new discussion material. Moreover, in the current manuscript, you do not show how fast the icebergs form in time, you only mention that they form "more quickly" once across-flow crevasses are introduced to the model (p8-l13). Videos in a supplementary material could be helpful to illustrate your point.

Due to the large storage requirement of the model's output files, we have stored only a limited amount of output data and we are not able to produce a video or similar as recommended. Instead, we have restricted our discussion of the results to the data that can be seen in Figures 6 & 7 (p9 l20-23).

**2.24**: p9, l17: "which are likely produced at the grounding line": Again, you'd need to show first through a sensitivity study that the underproduction of across-flow crevasses in your calving-front-HiDEM setup is not due to the choice in model parameters.

See discussion above regarding sensitivity testing. In addition to this, our models cannot discount that basal crevasses may also be formed on the ice shelf downstream of the grounding line. This is now acknowledged in the text (p11 l23-25).

**2.25**: Sect.4: Much of section 4 (p8, l20 to p9, l12-21) is discussion material, and would deserve a separate, earlier section. I suggest to then create the conclusions from a condensed version of the discussion and the outlook.

We have altered the paper's structure as suggested by the reviewer.

**2.26**: Figure 1: I suggest to add a scale ruler, a North arrow/grid lines, and an Antarctica location inset for orientation.

We have added an Antarctic location inset and grid lines for clarity. The Eastings/Northings grid used provides information equivalent to a north arrow and scale so we have not added these for simplicity.

**2.27**: Figure 2: Only little information is gained by showing two times the same satellite image. The reader will still get a feel for the crevassing pattern if image a was to be left out. Even more so, as the remaining image would be printed larger, and the crevassing pattern would be visible in more detail.

In response to other reviewer's comments, we have updated this figure to include model output from a later timestep, making the perpendicular-to –front fractures clearer to the reader. The format has also been updated to be consistent with Figure 6 to assist the reader with interpretation.

**2.28**: Figure 3: The main information of this figure should be the basal crevasses, which occupy only the bottom part of the plot. The most visible information is the colorful ice thickness, which stems from a data set not presented here. I'd therefore suggest to use a less dominating color map for the ice thickness (e.g. a gray scale). The crevasses (black) and the grounding line (dark gray) are hard to distinguish. I'd suggest using a brighter color for the crevasses.

Changes have been made to the figure as suggested.

**2.29**: Figure 4: Just a comment: The blue color of the vertical lines at the bottom associates them with the observed blue profile. I suggest changing their color to red, i.o. to associate them with the modeled profile. Please elaborate: does this figure show a steady state geometry?

The vertical lines at the bottom indicate the initial fractures used in the model set-up, which were directed by the observed geometry so we maintain this colour in order to make that clear to the reader. The figure does not show a steady state geometry, and we have altered the caption and relevant results section to make this clear (p9 l10).

**2.30**: Figure 5: The clarity of the figure could be enhanced by a) cropping the x and y axis so that the floating slab of ice is in the respective center of the subplot, b) dropping the 6-digit precision of the time which is superfluous, c) using a uniform format for the tick-labels of the colorbar, and d) including a y-axis.

We have updated the figure to include the requested changes.

**2.31**: Figure 6: This figure was confusing to me at first due to its grainy appearance. The meaning of the discontinuous horizontal black lines is not clear. The broken bonds between the ice bergs are hard to distinguish from only slightly damaged or undamaged ice. The Cartesian three-arrow orientation guide has no labels and could be left out. Additionally, x and y-labels, a scaleruler, gridlines and/or a north arrow would be helpful for orientation for the reader. Why are some of the areas up to the calving front white (and I assume not damaged?). I think it would be useful - for comparison of the crevasse pattern - to include the same figure next to it for the case without the added basal across-flow crevasses. Why not use the same layout at Figure 2b?

This figure is plotted so that, on a dark background representing water, all particle locations are marked by a white dot, smaller than the actual particle in the model. Since the glacier is modelled by discrete layers of particles the more shallow ice at the calving front turns grey as some water 'shines through', while the thicker ice is white. The boundary is distinct because the model is discrete. Where crevasses have formed, particles have moved away from each other, and they therefore show up as narrow black bands.

We have updated to figure caption to help the reader with interpretation of the figure, and added a north arrow and scale bar for context. We have also adapted Figure 2 to be compatible with this figure, which will also help to provide the reader with context.

**2.32**: Figure 7: The plot includes a fit to the iceberg size distribution. Are you trying to fit the observations or the model results? Are you able say anything about the exponent chosen for the fit, or conclude anything from it? Please discuss.

The theoretical distribution (area$^{-3/2}$) has not been fitted to the data, simply placed roughly overlaying them for comparison. From previous studies in fracture formation e.g. Åström (2006), the exponent of the distribution is the key variable in understanding the fracture processes involved, an exponent of -3/2 indicates brittle fracture while a higher exponent would indicate that grinding or crushing is occuring. The figure caption now makes this clearer, and the text has been expanded to include additional discussion of the topic (p10 l1-10).

Ref: Åström, J. A. Statistical models of brittle fragmentation. *Adv. Phys.* **55,** 247–278 (2006).

**3 Minor Corrections**

**3.1**: p5, l9: This sentence is incomplete. Add "it"? This has been altered (p6 l14)

**3.2**: p6, l4: 2000 km$^2$: For consistency, please use side lengths like for the second model domain. Model dimensions are now given in the model setup (p5 l21).

**3.3**: p9, l17: "This implies..": The sentence does not flow. Please rephrase. The sentence has been rewritten (p13 l3-5)

**Reviewer 2: J. Bassis**

**General Appreciation.**
This manuscript describes a suite of models used to describe the fracture patterns observed around Totten Ice Shelf, East Antarctica. The authors apply a discrete element models (HiDEM) a full Stokes model (Elmer/Ice) and a model meant to describe basal melt (the KPZ equation) to describe the formation and evolution of basal crevasses. The authors use the suite of models to infer that basal crevasses that initiate near the grounding zone advect to the calving front and are responsible for the checker board pattern of fractures observed near the calving front of the ice shelf. That basal crevasses that initiate upstream play an important role in calving closer to the calving front is not that surprising and is consistent with several prior studies. However, this is an interesting hypothesis—especially in light of the argument that the channel geometry is related to submarine melt—because it provides a direct connection between fractures/calving and ocean forcing and this study provides the first direct numerical simulation of some of the fundamental processes. Hence, the study has the potential to be improve our understanding of a set of processes that remain poorly understood.

Organizationally, I found the manuscript a bit confusing to follow. I think it would help readers significantly if the authors presented their hypotheses or questions earlier on in the manuscript. On my first reading, it seemed as though the authors were merely throwing a couple of unrelated models at a problem without any underlying reason or rational. It wasn't until near the end of the manuscript that I realized that the authors were testing the hypothesis that crevasses near the grounding line control the fracture pattern near the calving front. That the authors need the basal crevasses is interesting and should be emphasized to readers earlier on. Here, I think the authors could significantly improve the impact of the manuscript by adding section subtitles and sketching out the logical flow. In other words, it would be helpful to see that the authors are trying to explain the fracture distribution near the calving front of the ice shelf. This could either be presented as a fundamental question early on (i.e., basal crevasses are needed to explain the fracture pattern), or complexity could be introduced into the suite of models expressly to match this fracture pattern (i.e., the HiDEM simulation without basal crevasses could not represent the fracture pattern). Without this additional text, some of the models appear unmotivated. I do also have some questions about the models and assumptions, described in more detail below.

We have attempted to improve the clarity of the manuscript in reference to comments from both reviewers by:

- Explaining the hierarchy of models more clearly in the abstract.
- Expanding the motivation for the different models in the introduction (p3 l12-23).
- Increasing the discussion of model runs performed in the model setup (p5 l25 – p6 l14)
- Separating results and discussion into different sections.

**Questions about models.**
Overall, I think that the hierarchy of models is interesting. I do have a few questions about the models. For example:

-How many particles are used in the HiDEM model runs. What is the vertical number of particles? More details on the numerical setup would be helpful to readers.

Approximately 2.5 million particles are used for the grounding line model and 5 million for the calving front model. This corresponds to a minimum of 15 particles vertically for the grounding line model and 5 for the calving front model. This information has now been included in the text (p6 l3-6).

-How are the basal crevasses/channels used in the last simulation specified in HiDEM? Are these based on observations? Providing this data is critical.

To mimic the ice-base shape in Fig. 4, we cut-out a set of ~100 m deep and 0.5-1km wide embayments at the ice-base at ~0-2km intervals using trigonometric functions to positive power (e.g. $\sin(cx)^8\cos(bx)^2$). This is now described in the text (p6 l6-13).

-What ice temperature is used in the ELMER/ICE simulations? This should be important in determining the rate of crevasse widening. I'm also curious about the effect of the strain rate on widening. The strain rate is sufficient to allow crevasses to widen, but I would expect crevasses to actually close if the strain rate is smaller. This would be interesting to check.

The temperature of the ice is -9°C (p8 l6). We only consider strain thinning in this manuscript, as those are the conditions observed on the floating portion of the Totten Ice Shelf. The reviewer is correct that if the thinning were less then the basal crevasses would be likely to close, and this may be considered in future simulations on different ice shelves. This has now been acknowledged in the discussion of the results (p12 l1-2).

-Similarly, I would have liked to see some more quantitative comparisons between observations and data if feasible. For instance, how wide are the fractures predicted by HiDEM and do they penetrate the entire ice thickness? It is difficult to detect if fractures penetrate the entire ice thickness using surface observations, but one could estimate the width of fractures (based on pixels) and compare this to the predicted width of fractures from HiDEM. In fact, it would be really nice to have a statistical comparison between length, width and orientation of fractures predicted by HiDEM and those observed in the images. This would significantly bolster the authors claims and avoid the need to rely on qualitative inspection of figures to determine how well the model is performing. This is important for a range of reasons, but it is important to recognize that the iceberg size distribution is modified by interaction with the ocean (bergs melt, erode, tip over, etc.) and hence the comparison between berg sizes is not necessarily a one-to-one comparison with bergs that detach. Fracture distributions should be more straightforward to map and compare.

We have struggled to find a robust way to quantitatively compare the modelled and observed fractures for a number of reasons. Firstly, the dimensions of observed fractures are hard to determine accurately because of the thick snow and firn layer on the ice shelf – this snow blurs the edges of fractures, making robust measurements of length and width extremely difficult. For full thickness rifts at the edges of the ice shelf, the evolution of their width will also be affected by an ice mélange (of frozen ice debris and sea ice) which we cannot adequately represent in the model, meaning that the modelled rifts will likely develop further than in the observed shelf.

The orientation of fractures is the most promising method for comparing the two sets of fractures. We have digitized both the observed and modelled fractures, and created a dataset of orientations of each line segment. The results have now been included in Supplementary material, and the two orientation distributions are shown in Supp. Fig. 2. The highest frequency in the observed fractures is in the across-flow direction, while this orientation is much less frequent in the modelled fractures supporting our conclusion that these fractures are not produced locally.

We have retained our discussion of the iceberg size distributions observed and produced by the model. The reviewer is correct that a number of processes will affect the size of icebergs after they detach form the shelf, but previous work has shown that the observed iceberg size distribution around Antarctica is actually very close to that expected from pure brittle fracture (Tournadre et al, 2016). The iceberg size distribution is useful because it provides information on the fracture processes occurring in the model, which cannot be extracted as easily from the fracture distribution alone. We have expanded our discussion around this point in the text (p10 l1-10).

Ref: Tournadre, J., Bouhier, N., Girard-Ardhuin, F. and Remy, F.: Antarctic icebergs distributions 1992–2014, J. Geophys. Res. Ocean., 121, 327–349, doi:10.1002/2015JC010969, 2016.

A few other quantitative questions I had about data included: How many total icebergs were detached in 2009, 2010 and 2011 respectively and how many icebergs in each size class were detected? What time of year are these images? Is there a bias associated with the time of year? Furthermore, it is possible that the distribution of icebergs/ calving rate is not stationary stochastic. If this is the case, then the distribution of icebergs would not be stationary stochastic, as the authors assume.

To obtain the current iceberg size distribution for Totten Ice Shelf, icebergs were manually digitised from three Landsat 7 images from 23/11/2009, 26/11/2010 and 29/11/2011. The dataset consists of 370 icebergs, broken down by year: 83 (2009), 148 (2010), 139 (2011). These years cover a wide range of average calving rate, as measured by Dr. Yan Liu of Beijing Normal University, leading us to consider that we have sampled the full range of calving behaviour of the ice shelf. Broken down by year, the three iceberg size distributions look similar therefore we consider this to be a robust test of the model. Our distribution is also consistent with pan-Antarctic icebergs distributions 1992–2014, measured by Tournadre et al. (2016). This is now discussed in detail in the new addition of supplementary material to the paper.

**KPZ equation** The use of the KPZ equation in this context is interesting and novel. I have played around with it a few times for various things. My understanding of the usage of the KPZ equation here is the $v$, $\lambda$ and $F$ (which includes two terms to allow for a linearly varying profile) are all adjusted to match observations. This means that the authors have 4 free

parameters to adjust. From looking at Figure 4, it also looks like the position, width and initial depth of the crevasses are all also determined as part of the solution. There is an old saying that given three parameters you can fit an elephant. Given 5, you can make its trunk wiggle. Although, I find this a bit extreme, I do think it is worth discussion the robustness of the results in the face of 7 unconstrained parameters. This should start by listing the numerical values of parameters used. Are results sensitive to order of magnitude changes in parameters? Factors of two changes? I also have questions about the fact that, as far as I can tell the melt rate ($F$) is not related to ocean forcing at all. I guess this is just an empirical constant, but the KPZ model is of limited utility if we cannot relate it to ocean forcing. At the very least, we have observations of the melt rate and hence, can constrain the average value of the melt forcing term. Similarly, the time at which the simulation result is obtained is not given. How long does it take for these channels to evolve and how different do the channels look at different instances of time. This is crucial because the KPZ equation does not necessarily admit a steady-state—similarity solutions exist that allow for continuous variation. If the solution is steady-state, then this should be declared along with how long it takes the simulation to arrive at steady-state.

**General comment on the KPZ approach:**

The use of the KPZ-equation in this context is, at best, a crude phenomenological description of the way basal crevasses are formed and developed on the Totten Ice Shelf. In our opinion the interesting question related to this approach is whether a simple and straightforward implementation of the KPZ-equation can largely reproduce the basal shape or not. The model is much too crude to learn new information from detailed exploration of the model details.

As I mentioned, I think the use of the KPZ equation here is interesting and novel. However, it appears as though the KPZ equation as listed in Equation (4) has an error. The actual equation should read:

$$\frac{\partial h}{\partial t} + u.\nabla h = \nu\nabla^2 h + \frac{\lambda}{2}[\nabla h]^2 + F + \eta\,,$$

where $u$ is the velocity along the interface. There is no physical reason to omit the advection term. Typically, the KPZ equation is used for materials where the interface does not have a background velocity (as in the refs cited) and hence, one can replace the material derivative with the partial derivative with respect to time, as the authors seem to have done. In this case, however, the interface is moving with the background velocity of the ice and there is a gradient in the velocity. If the "fractures" are indeed enlarged by a constant melt rate, then including the advection term should allow the authors to reproduce the spatial pattern of fractures using a constant melt rate $F$ instead of a linearly varying melt rate. Now, because velocity is not Galilean invariant one need only specify the difference in velocity across the domain. One could do this in a consistent manner with the Elmer/Ice model by using the same strain rate across the domain.

We have chosen to treat the x-coordinate in KPZ equation as a Lagrangian coordinate (x'), such that it is attached to the ice moving at ~1 km a$^{-1}$. This is a rough simplification of the region covered by the flight path shown in Fig. 4, where the velocities range from roughly 1500 m a$^{-1}$ upstream to 1700 m a$^{-1}$ downstream. With the coordinate transformation between our coordinate system (x') and the one suggest by the reviewer (x),  x' = x- u*t, the

advection vanishes (dx/dt = dx'/dt - u = 0). What needs to be taken into account then is that the ice further downstream has been exposed to melt for a longer time period. With a constant melt rate of approximately 2 m a$^{-1}$ and ice speed of 1 km a$^{-1}$, we get an interface slope of ~10$^{-3}$. This is now laid out more clearly in the text, with our justification for treating the front as stationary (p6 l20-25) and a description of the Lagrangian coordinate system used (p7 l1-3).

Given the number of simplifications made by using the KPZ equation (particularly the homogeneous melt rate), we consider the order of magnitude to be the important factor in interpreting the results. It is largely meaningless to try to fit exact values for a crude phenomenological model like this one. The free parameters $F$, $\lambda$ and $v$ are melt rates and should be of order 1 m a$^{-1}$ according to ocean modelling (Gwyther et al., 2014). The only adjustable input in the model is therefore the initial line-fracture configurations. The figure below demonstrates the effect of varying the melt rate used for $F$, $\lambda$ and $v$ – the size (both horizontally and vertically) of the features produced and the slope of the shelf are affected by the choice of melt rate, while their location and relative magnitude are fixed by the choice of initial line fractures (η).

[Figure]

What the exercise leading to Fig.4 shows is that it is possible to choose a suitable starting configuration of line-cracks, apply KPZ-dynamics with reasonable parameter values and arrive at an ice-surface and ice-base that match observations. We have tested the KPZ equation with different setups to test the robustness of the result in Fig.4. For example we tested Eq. (5) (previously Eq. 4 in the old manuscript) with and without random fluctuations. In both cases it is possible to arrive at results similar to Fig.4 with minor (i.e. within a factor ~2) adjustments of F, λ and v.

There are other, subtle, issues associated with the use of the KPZ equation in this context. One prominent assumption of the KPZ equation is that the erosion of the interface does not depend on the absolute elevation of the interface. This is the reason why only scalar powers of rh appear in the expansion. In the case of submarine melt, however, this assumption is violated because of the pressure dependence of the melting point. Erosion of ice should occur more rapidly deep beneath the surface of the ocean and erosion (or even marine ice deposition) should occur closer to the ocean surface. As a consequence, I'm not even sure that the *sign* of the $[\nabla h]^2$ term is right. Expanding on this point, the KPZ equation is useful for describing interface evolution because it represents a large universality class of processes that, irrespective to process details, ultimately behave like the KPZ equation. The universality class, however, is not infinitely large. Here, I think the authors should attempt to establish whether submarine melt actually resides in the broader universality class exemplified by the KPZ equation. Failing this, I would really like more detail on **why** the authors think the KPZ equation is an adequate description of the submarine melting process and the role of the pressure dependence of the melt point in this argument.

It would indeed be nice to establish whether the interface created by basal melt actually belongs to the KPZ universality class. To do this properly it would demand enough data to perform a scaling analysis and compute scaling exponents. We do not currently have enough data to do this. The only thing we attempt in this paper is to test the hypothesis that KPZ actually can describe the interface, and it can as evidenced by Fig.4. One can, of course, speculate slightly about what would eventually be the outcome of the melt-pressure dependence with a melt-freeze boundary. For long time periods (compared to glacier thickness/melt-rate) the entire interface should then asymptotically reach the equilibrium line between melt and freeze. Assuming melt-rate only has a simple height dependence, it would induce a flat interface. This is obviously not case, indicating that the ocean water is warm enough to drive the interface. As long as this happens, the KPZ analysis seems to capture the dynamics reasonably well. This is not very surprising if the phase-diagram for (fresh) water and ice is investigated. The phase-transition between ice and water: It is practically a constant (0°C) up to pressures of 10MPa, which corresponds to a water depth of about 1000 m. Only for deeper water does the melting point decrease significantly, making this dependence largely irrelevant for this particular case.

Finally, I was confused about the role of the Elmer/Ice and KPZ simulations of basal channel evolution. Why not incorporate the KPZ equation into the ELMER/ICE simulations so that the authors can self-consistently simulate the strain induced widening and erosion? If this isn't feasible, then what are we supposed to learn from the two simulations? As far as I can tell, the Elmer/Ice simulation tells us that crevasses widen and become shallower over time, but the Elmer/Ice simulations omit ocean forcing. In contrast, the KPZ simulations tell us that crevasses become wider and deeper over time, but this simulation omits any ice dynamics (included advection!). Do these effects cancel out? Does one dominate? Do we need both or can we omit one of them? I do wonder if the widening by ice dynamics alone is sufficient to explain the width of observed basal crevasses. If it is not sufficient, then this argues that erosion by submarine melt is necessary.

We do not quite understand what the reviewer means by incorporating the KPZ into Elmer/Ice. We guess he means introducing thermodynamics and thereby melting at the interface. These would be very interesting, but rather demanding computations and we do

not currently have the resources to perform them. What we can say is that, as the reviewer points out, both KPZ and Elmer predicts widening of the crevasses. This is consistent with our hypothesis that the crevasses originate from thin fractures in the grounding zone.

The reviewer points out that flow will make the crevasses shallower, while melt will make them deeper. Unfortunately, we cannot use the models presented here to make any conclusions from this about the relative importance of the two processes. The KPZ has assumed a homogeneous melt rate, while previous modelling work has suggested that melt will be highest along the walls of basal fractures while refreezing may occur at the peak (Jordan et al., 2014; Khazendar and Jenkins, 2003). Meanwhile, while our simulation suggests that longitudinal stretching may make basal crevasses shallower, other modelling work has suggested that the concentration of tensile stresses in the thin ice above the crevasse may enhance further brittle fragmentation allowing the basal crevasse to grow (Bassis and Ma, 2015). To properly assess how these processes would interact would require a combined ocean, ice flow and fracture model. We have extended our discussion in the text to include these topics (p11 l17 – p12 l10).

**Detailed comments:**
Page 3: "We also use a continuum ice flow model and a stochastic equation describing fracture development" This is unclear. It seems as though the KPZ equation is used to model the evolution of the interface in response to melt. The KPZ equation is indeed stochastic. However, the authors omit the stochastic terms, at which point version of the KPZ that they use is entirely deterministic . . . Also, the authors claim $\eta$ is a stochastic term, but then they set it equal to the initial width/length of basal crevasses. This seems to confuse initial conditions with the stochastic forcing. Typically, the stochastic term in the KPZ emerges as a white noise Gaussian process, which is quite different from the assumptions made here. It might be less confusing to separate this into an initial condition and noise term with the noise term set to zero.

This is a minor point given the fact that the stochastic term is set to zero, but stochastic partial differential equations fall into two categories the so-called Ito and Stratonovich interpretations. These two interpretations are subtly different in important ways, but necessary to place stochastic differential equations on firm mathematical footing. What interpretation is assumed here?

This is not really a topic for a publication in the Cryosphere, and with a stochastic term set to zero, the division into Ito and Stratonovich become irrelevant. Where we used a non-zero η, we used the Ito formulation. It seems to be the consensus that for climate/ocean/etc models Stratonovich would be preferable. However, in our case, we cannot even make a proper interpretation of what would be the right form of η, and the question is superceded by the other simplifications used in setting up the model. We have rewritten the original sentence to make its meaning clearer (p3 l19).

Page 4: How big of a difference does it make to results if a different percentage of bonds is broken? Say 0.1% or 10%?

The proportion of bonds broken has little effect on the results, so long as it is below the percolation limit (see Fracture Mechanics: With an Introduction to Micromechanics by

Dietmar Gross and Thomas Seelig for a discussion of the application of percolation theory to fracture propagation and material failure). The value we have chosen is well below the percolation limit.

Page 4 last sentence after equation 3. The constant "c" is apparently dimensionless (at least no units are given) and thus friction does not have the correct units. Either "c" should have units or units of friction needs to be more carefully explained (have the equations themselves been non-dimensionalized)?

The units were overlooked and have now been added (p5 l22).

Figure 2: Am I missing something? Fractures look like the run both parallel and perpendicular to the front?

We have updated the figure to use fracture data from later timestep, which we believe makes the pattern clearer to the reader. We have also addressed the reviewer's underlying concern about the fracture orientation by further analysis described above.

Figure 3 is hard to decipher. Is it possible to show Figure 3 in the same style of as Figure 2 so that we can see where the crevasses are located on an image?

High snowfall in this region of the ice shelf means that the surface has very few features visible in satellite imagery. We have retained the ice thickness as background, but plotted in grayscale as suggested by reviewer #1 to improve clarity.

Figure 6 is very pixelated (maybe a conversion issue?) and hard to interpret.

The figure file will be provided in highest possible resolution for paper production to solve any pixilation issues. We have also updated to figure caption to help the reader with interpretation of the figure.

Figure 7. It doesn't look like the simulation obeys a -3/2 scaling law. According to the red curve, the computation predicts more icebergs at low iceberg area and not enough at large iceberg area. The observations are really only power-law over about an order of magnitude. Fitting power-laws to such a limited data range is fraught with peril. This maybe an example where it is better to avoid a logarithmic plot AND to include error bars in the observations. How many icebergs are there with sizes close to $10^6$ m$^2$? Is this actually statistically significant?

For this figure we did not fit a power law (hence the lack of error bars). Instead, we overlaid a theoretical distribution based on brittle fragmentation. The reviewer is correct that the simulation does not obey the -3/2 scaling law perfectly. The origin of this behaviour originates in two different mechanisms of fracture: pure tensile fracture, and crushing small debris in shear-bands. This is an extensive topic which we will address in more detail in a separate paper. The discussion of this was lacking in both the text and the figure caption, and has now been expanded (see revised figure caption and p10 l1-10 in updated text).